# TOWARDS THE DETECTION OF DIFFUSION MODEL DEEPFAKES

## ABSTRACT

Diffusion models (DMs) have recently emerged as a promising method in image synthesis. They have surpassed generative adversarial networks (GANs) in both diversity and quality, and have achieved impressive results in text-to-image and image-to-image modeling. However, to date, only little attention has been paid to the *detection* of DM-generated images, which is critical to prevent adverse impacts on our society. Although prior work has shown that GAN-generated images can be reliably detected using automated methods, it is unclear whether the same methods are effective against DMs. In this work, we address this challenge and take a first look at detecting DM-generated images. We approach the problem from two different angles: First, we evaluate the performance of state-of-the-art detectors on a variety of DMs. Second, we analyze DM-generated images in the frequency domain and study different factors that influence the spectral properties of these images. Most importantly, we demonstrate that GANs and DMs produce images with different characteristics, which requires adaptation of existing classifiers to ensure reliable detection. We believe this work provides the foundation and starting point for further research to detect DM deepfakes effectively.

## 1 INTRODUCTION

In the recent past, diffusion models (DMs) have shown a lot of promise as a method for synthesizing images. Such models provide better (or at least similar) performance compared to generative adversarial networks (GANs) and allow powerful text-to-image models such as DALL-E 2 (Ramesh et al., 2022), Imagen (Saharia et al., 2022), and Stable Diffusion (Rombach et al., 2022). Advances in image synthesis have resulted in very high-quality images being generated, and humans can hardly tell if a given picture is an actual or artificially generated image (so-called *deepfake*) (Nightingale & Farid, 2022). This progress has many implications in practice and poses a danger to our digital society: Deepfakes can be used for disinformation campaigns, as such images appear particularly credible due to their sensory comprehensibility. Disinformation aims to discredit opponents in public perception, to create sentiment for or against certain social groups, and thus influence public opinion. In their effect, deepfakes lead to an erosion of trust in institutions or individuals, support conspiracy theories, and promote a fundamental political camp formation. Despite the importance of this topic, there is only a limited amount of research on effective deepfake detection. Previous work on the detection of GAN-generated images (e.g., Wang et al. (2020), Gragnaniello et al. (2021), and Mandelli et al. (2022a)) showed promising results, but it remains unclear if any of these methods can be applied to DM-generated images.

In this paper, we present the first look at detection methods for DM-generated media. We tackle the problem from two different angles. On the one hand, we investigate whether DM-generated images can be effectively detected by existing methods that claim to be universal. We study ten models in total, five GANs and five DMs. We find that existing detection methods suffer from severe performance degradation when applied on DM-generated images, with the area under the receiver operating characteristic curve (AUROC) metric dropping by 15.2% on average compared to GANs. These results hint at a structural difference between synthetic images generated by GANs and DMs. We show that existing detection methods can be improved by fine-tuning, which makes detection almost perfect. However, our results also suggest that recognizing DM-generated images is a more difficult task than recognizing GAN images.

On the other hand, we analyze DM-generated images in the frequency domain and compare them to GAN-generated images. Although DMs do not exhibit strong frequency artifacts compared to GANs, their spectrum deviates from real images. We hypothesize that discrepancies in spectral properties are a possible reason for our identified differences. Therefore, we analyze the spectral properties of DM- and GAN-generated images in detail and find that high frequencies are systematically mismatched. Further analysis suggests that too little weight is given to these frequencies during training due to the choice of the training objective. We believe that our results provide the foundation for further research on the effective detection of deepfakes generated by DMs.

## 2 RELATED WORK

**Universal Fake Image Detection**   While in recent years a variety of successful methods to detect artificially generated images has been proposed (Verdoliva, 2020), generalization to unseen data remains a challenging task (Cozzolino et al., 2019). Constructing an effective classifier for a specific generator is considered straightforward, which is why more research effort is put into designing universal detectors (Xuan et al., 2019; Chai et al., 2020; Wang et al., 2020; Cozzolino et al., 2021; Gragnaniello et al., 2021; Girish et al., 2021; Mandelli et al., 2022a). This is especially important in the context of deceptive media, since new generative models emerge on a frequent basis and manually updating detectors is too slow to stop the propagation of harmful contents.

**Frequency-Based Deepfake Detection**   Zhang et al. (2019) were the first to demonstrate that the spectrum of GAN-generated images contains visible artifacts in the form of a periodic, grid-like pattern due to transposed convolution operations. These findings were later reproduced by Wang et al. (2020) and extended to the discrete cosine transform (DCT) by Frank et al. (2020). Another characteristic was discovered by Durall et al. (2020), who showed that GANs are unable to correctly reproduce the spectral distribution of the training data. In particular, generated images contain increased magnitudes at high frequencies. While several works attribute these spectral discrepancies to transposed convolutions (Zhang et al., 2019; Durall et al., 2020) or, more general, up-sampling operations (Frank et al., 2020; Chandrasegaran et al., 2021), no consensus on their origin has yet been reached. Some works explain them by the spectral bias of convolution layers due to linear dependencies (Dzanic et al., 2020; Khayatkhoei & Elgammal, 2022), while others suggest the discriminator is not able to provide an accurate training signal (Chen et al., 2021; Schwarz et al., 2021).

**Detection of DM-Generated Images**   Despite the massive attention from the scientific community and beyond, DMs have not yet been studied from the perspective of image forensics. A very specific use case is considered by Mandelli et al. (2022b), where the authors evaluate methods for detecting western blot images synthesized by different models, including DDPM (Ho et al., 2020). More related to our analysis is the work of Wolter et al. (2022), which proposes to detect generated images based on their wavelet-packet representation, combining features from pixel- and frequency space. While the focus lies on GAN-generated images, they demonstrate that the images generated by ADM (Dhariwal & Nichol, 2021) can be detected using their approach. They also report that the classifier "appears to focus on the highest frequency packet", which is consistent with our findings.

**Frequency Considerations in DMs**   Both Kingma et al. (2021) and Song et al. (2022b) experiment with adding Fourier features to improve learning of high-frequency content, the former reporting it leads to much better likelihoods. Another interesting observation is made by Rissanen et al. (2022) who analyze the generative process of diffusion models in the frequency domain. They state that diffusion models have an inductive bias according to which, during the reverse process, higher frequencies are added to existing lower frequencies.

## 3 BACKGROUND ON DMS

DMs were first proposed by Sohl-Dickstein et al. (2015) and later advanced by Ho et al. (2020), who also pointed out the connections between DMs and score-based generative models (Song & Ermon, 2019; 2020; Song et al., 2022b). Since then, numerous modifications and improvements have been proposed, leading to higher perceptual quality (Nichol & Dhariwal, 2021; Dhariwal & Nichol, 2021; Choi et al., 2022; Rombach et al., 2022) and increased sampling speed (Song et al.,

2022a; Liu et al., 2022; Salimans & Ho, 2022; Xiao et al., 2022). In short, a DM models a data distribution by gradually disturbing a sample from this distribution and then learning to reverse this diffusion process. In the diffusion (or forward) process, a sample $\mathbf{x}_0$ (an image in most applications) is repeatedly corrupted by adding Gaussian noise in sequential steps $t = 1, \ldots, T$:

$$q(\mathbf{x}_t|\mathbf{x}_{t-1}) = \mathcal{N}(\mathbf{x}_t; \sqrt{1 - \beta_t}\mathbf{x}_{t-1}, \beta_t \mathbf{I}) \ , \tag{1}$$

where $\beta_t$ defines a noise schedule. With $\alpha_t := 1 - \beta_t$ and $\bar{\alpha}_t := \prod_{s=1}^{t}(1 - \beta_t)$, we can directly create a corrupted sample $x_t$ via

$$q(\mathbf{x}_t|\mathbf{x}_0) = \mathcal{N}(\mathbf{x}_t; \sqrt{\bar{\alpha}_t}\mathbf{x}_0, (1 - \bar{\alpha}_t)\mathbf{I}) \ . \tag{2}$$

During the denoising (or reverse) process, we aim to sample from $q(\mathbf{x}_{t-1}|\mathbf{x}_t)$ to obtain a clean image given $\mathbf{x}_T$. However, since $q(\mathbf{x}_{t-1}|\mathbf{x}_t)$ is intractable as it depends on the entire underlying data distribution, it is approximated by a deep neural network. More formally, $q(\mathbf{x}_{t-1}|\mathbf{x}_t)$ is approximated by

$$p_\theta(\mathbf{x}_{t-1}|\mathbf{x}_t) = \mathcal{N}(\mathbf{x}_{t-1}; \mu_\theta(\mathbf{x}_t, t), \Sigma_\theta(\mathbf{x}_t, t)) \ , \tag{3}$$

where mean $\mu_\theta$ and covariance $\Sigma_\theta$ are given by the output of the model (or the latter is set to a constant as proposed by Ho et al. (2020)). Predicting the mean of the denoised sample $\mu_\theta(\mathbf{x}_t, t)$ is conceptually equivalent to predicting the noise that should be removed, denoted by $\epsilon_\theta(\mathbf{x}_t, t)$. Predominantly, the latter approach is implemented (e.g., by Ho et al. (2020); Dhariwal & Nichol (2021)) such that training a DM boils down to minimizing a (weighted) mean-squared error (MSE) $\|\epsilon - \epsilon_\theta(\mathbf{x}_t, t)\|^2$ between the true and predicted noise.

## 4 GENERATIVE MODELS UNDER EVALUATION

In contrast to previous work, we evaluate all generative models trained on the same dataset. This avoids any biases due to the complexity of the dataset when estimating the accuracy of deep fake detection methods, e.g., a high detection accuracy for a model trained on a complex dataset that cannot be modeled as well as a simpler dataset. It also provides fair conditions for evaluating a fine-tuned detector's ability to generalize to unseen data. In addition, having only a single dataset is beneficial for frequency analysis. Although natural images have been shown to have similar power spectra across different domains (Burton & Moorhead, 1987; Field, 1987; 1999; Tolhurst et al., 1992; Torralba & Oliva, 2003), a model may be able to reproduce the frequencies of one dataset better than another.

We evaluate models trained on LSUN Bedroom (Yu et al., 2016) due to the availability of pre-trained models and/or samples for many state-of-the-art GANs and DMs. This dataset also has the advantage that no resizing operations (besides cropping)[1] are required, which are a potential source of error due to faulty implementations (Parmar et al., 2022) or a mismatch between pre-processing for real and generated images (Chai et al., 2020). To demonstrate that our findings are not dataset specific, we show results on additional data in Appendix D.

An overview of the models under evaluation is shown in Table 1, and example images are given in Appendix A. We consider data from

Table 1: **Models evaluated in this work.** Fréchet inception distances (FIDs) (Heusel et al., 2017) on LSUN Bedroom 256×256 are taken from the original publications and from Dhariwal & Nichol (2021) in the case of IDDPM. A lower FID corresponds to higher image quality.

| Model Class | Method | Publication | FID |
|---|---|---|---|
| GAN | ProGAN | Karras et al. (2018) | 8.34 |
| | StyleGAN | Karras et al. (2019) | 2.65 |
| | ProjectedGAN | Sauer et al. (2021) | 1.52 |
| | Diff-StyleGAN2 | Wang et al. (2022) | 3.65 |
| | Diff-ProjectedGAN | Wang et al. (2022) | 1.43 |
| DM | DDPM | Ho et al. (2020) | 6.36 |
| | IDDPM | Nichol & Dhariwal (2021) | 4.24 |
| | ADM | Dhariwal & Nichol (2021) | 1.90 |
| | PNDM | Liu et al. (2022) | 5.68 |
| | LDM | Rombach et al. (2022) | 2.95 |

ten models in total, five GANs and five DMs. This includes the seminal models ProGAN and StyleGAN, as well as the more recent ProjectedGAN. Note that the recently proposed methods, Diff(usion)-StyleGAN2 and Diff(usion)-ProjectedGAN (the current state of the art on LSUN Bedroom) use a forward diffusion process to optimize GAN training, but this does not change the GAN model architecture. From the class of DMs, we consider the original DDPM, its successor IDDPM,

---

[1] All images have the smaller dimension equal to 256 pixels.

and ADM, the latter outperforming several GANs with an FID of 1.90 on LSUN Bedroom. PNDM optimizes the sampling process by the factor of 20 using pseudo numerical methods, which can be applied to existing pre-trained DMs. Lastly, LDM uses an adversarially trained autoencoder that transforms an image from the pixel space to a latent space. Training the DM in this more suitable latent space reduces the computational complexity and therefore enables training on higher resolutions. The success of this approach is underpinned by the recent publication of Stable Diffusion[2], a powerful and publicly available text-to-image model based on LDM. An interesting detail is that the autoencoder used to transform images is very similar to a VQGAN (Esser et al., 2021).

All samples have dimensions $256 \times 256$ and are either directly downloaded or generated using code and pre-trained models provided by the original publications. More detailed descriptions are given in Appendix A. For each model, we collect 50k samples in total from which 39k are used for training, 1k for validation, and 10k for testing and frequency analysis, if not stated otherwise.

# 5 DETECTION OF DM-GENERATED IMAGES

We now investigate the effectiveness of state-of-the-art fake image detectors on DM-generated images. First, we analyze the performance of off-the-shelf pre-trained models which are highly capable of detecting images generated by GANs, and then perform experiments with models fine-tuned on DM-generated images. We also show example images considered more or less "fake" based on the detector's output in Appendix B.5.

In this work, we evaluate three state-of-the-art detection methods by Wang et al. (2020), Gragnaniello et al. (2021), and Mandelli et al. (2022a). A description of these detectors is provided in Appendix B.1. All three claim to perform well on unseen data, but it is unclear whether this holds for DM-generated images as well.

The performance of the analyzed classifiers is estimated in terms of the widely used AUROC. As pointed out by Cozzolino et al. (2021), however, the AUROC tends to be overly optimistic since it captures merely the potential of a classifier, but the optimal threshold is usually unknown. Therefore, we adopt the use of the probability of detection at a fixed false alarm rate (Pd@FAR) as an additional metric, which is given as the true positive rate at a fixed false positive rate. Intuitively, this corresponds to picking the y-coordinate of the ROC curve given an x-coordinate. This metric is a valid choice for realistic scenarios such as large-scale content filtering on social media, where only a certain amount of false positives is tolerable. We consider FARs of 5% and 1%.

## 5.1 CAN WE USE STATE-OF-THE-ART DETECTORS FOR DM-GENERATED IMAGES?

Our first analysis aims to answer the question whether DM-generated images are sufficiently similar, with respect to detectable traces, to those generated by GANs.

Table 2: **Detection performance of pre-trained universal detectors.** For Wang et al. (2020) and Gragnaniello et al. (2021), we consider two different variants, respectively. The best score (determined by the highest Pd@1%) for each generator is highlighted in **bold**. We also report average scores per detector and model class in gray.

| AUROC / Pd@5% / Pd@1% | Wang et al. (2020) | | Gragnaniello et al. (2021) | | Mandelli et al. (2022a) |
|---|---|---|---|---|---|
| | Blur+JPEG (0.5) | Blur+JPEG (0.1) | ProGAN | StyleGAN2 | |
| ProGAN | **100.0 / 100.0 / 100.0** | **100.0 / 100.0 / 100.0** | **100.0 / 100.0 / 100.0** | **100.0 / 100.0 / 100.0** | 91.2 / 54.6 / 27.5 |
| StyleGAN | 98.7 / 93.7 / 81.4 | 99.0 / 95.5 / 84.4 | **100.0 / 100.0 / 100.0** | **100.0 / 100.0 / 100.0** | 89.6 / 43.6 / 14.7 |
| ProjectedGAN | 94.8 / 73.8 / 49.1 | 90.9 / 61.8 / 34.5 | **100.0 / 99.9 / 99.3** | 99.9 / 99.6 / 97.8 | 59.4 / 8.4 / 2.4 |
| Diff-StyleGAN2 | 99.9 / 99.6 / 97.9 | 100.0 / 99.9 / 99.3 | **100.0 / 100.0 / 100.0** | **100.0 / 100.0 / 100.0** | 100.0 / 100.0 / 99.9 |
| Diff-ProjectedGAN | 93.8 / 69.5 / 43.3 | 88.8 / 54.6 / 27.2 | **99.9 / 99.9 / 99.2** | 99.8 / 99.6 / 96.6 | 62.1 / 10.5 / 2.8 |
| Average | 97.4 / 87.3 / 74.3 | 95.7 / 82.4 / 69.1 | 100.0 / 100.0 / 99.7 | 99.9 / 99.8 / 98.9 | 80.4 / 43.4 / 29.5 |
| DDPM | 85.2 / 37.8 / 14.2 | 80.8 / 29.6 / 9.3 | **96.5 / 79.4 / 39.1** | 95.1 / 69.5 / 30.7 | 57.4 / 3.8 / 0.6 |
| IDDPM | 81.6 / 30.6 / 10.6 | 79.9 / 27.6 / 7.8 | **94.3 / 64.8 / 25.7** | 92.8 / 58.0 / 21.2 | 62.9 / 7.0 / 1.3 |
| ADM | 68.3 / 13.2 / 3.4 | 68.8 / 14.1 / 4.0 | **77.8 / 20.7 / 5.2** | 70.6 / 13.0 / 2.5 | 60.5 / 8.2 / 1.8 |
| PNDM | 79.0 / 27.5 / 9.2 | 75.5 / 22.6 / 6.3 | 91.6 / 52.0 / 16.6 | **91.5 / 53.9 / 22.2** | 71.6 / 15.4 / 4.0 |
| LDM | 78.7 / 24.7 / 7.4 | 77.7 / 24.3 / 6.9 | 96.7 / 79.9 / 42.1 | **97.0 / 81.8 / 48.9** | 54.8 / 7.7 / 2.1 |
| Average | 78.6 / 26.8 / 9.0 | 76.6 / 23.7 / 6.8 | 91.4 / 59.3 / 25.7 | 89.4 / 55.2 / 25.1 | 61.4 / 8.4 / 2.0 |

---

[2] https://github.com/CompVis/stable-diffusion

Using the dataset introduced in Section 4, we analyze the performance of the three state-of-the-art detectors. We use pre-trained models provided by the authors without any modifications and compute the evaluation metrics using 10k real and 10k generated images. The results are given in Table 2. For GAN-generated images, both classifiers provided by Gragnaniello et al. (2021) achieve almost perfect detection results across all five generators, even w.r.t. the challenging Pd@1% metric. For images generated by ProjectedGAN and Diff-ProjectedGAN, the variant trained on ProGAN images performs slightly better. The reduced scores for these two models could in part be explained by the high image quality indicated by low FIDs (see Table 1). In comparison, the performance of the classifier provided by Wang et al. (2020) is reduced (except for ProGAN, on which the model was trained), which confirms the findings of Gragnaniello et al. (2021). Similar to Gragnaniello et al. (2021), ProjectedGAN and Diff-ProjectedGAN achieve the lowest scores, with Pd@1% dropping below 50%. Except for Diff-StyleGAN2, Mandelli et al. (2022a) achieves significantly worse results than its contestants. A possible explanation could be that StyleGAN2 images make up a large portion of the training data for this classifier.

For images generated by DMs, we observe a sharp deterioration for all detectors, with AUROC dropping by 15.2% on average in comparison to GANs. The difference becomes even more severe when looking at the more realistic Pd@FAR metrics. While the best model by Gragnaniello et al. (2021) (trained on ProGAN images) achieves a Pd@5% of 100.0% and Pd@1% of 99.7% on GAN-generated images, the scores drop by 40.7% and 74.0% for DMs, respectively. Given a setting where a low FAR is required, we find that none of the detectors under evaluation are usable. Among the five DMs, ADM generated the images most difficult to detect, which corresponds to a low FID. However, images from LDM, which achieves the second lowest FID, can be detected relatively effectively. A potential reason for this could be the adversarial training procedure of LDM (see Section 4), which could result in properties similar to those of GAN-generated images.

Overall, we observe that pre-trained universal detectors suffer from a severe performance drop when applied to DM-generated images (compared to GANs). We conclude that there exists a structural difference between GAN- and DM-generated images that requires the adaptation of existing methods or the development of novel methods to ensure the reliable detection of synthesized images. In Appendix B.4 we also experiment with common image perturbations and show that these have a stronger effect on DM-generated images.

## 5.2 CAN WE IMPROVE DETECTION BY FINE-TUNING?

Given the findings presented above, the question remains whether DMs evade detection in principle, or whether the detection performance can be increased by fine-tuning.[3] We select Wang et al. (2020) as a baseline since the original training code is available[4], which guarantees using the same training procedure. Furthermore, we choose the configuration Blur+JPEG (0.5) as it yields slightly better scores on average, and use the validation set for adapting the learning rate and performing early stopping. Besides fine-tuning a model on data from each generator, we also consider three aggregated settings in which we fine-tuned on all images generated by GANs, DMs, and both, respectively.

The results are depicted in Figure 1. A model which is fine-tuned on images from a specific DM achieves near-perfect scores, with PD@1% ranging from 97.7% for ADM to 100.0% for PNDM and LDM. To a limited extent, models are also capable of generalizing to data from other DMs, indicating common, detectable characteristics. Interestingly, detectors fine-tuned on images from DMs are significantly more successful in detecting GAN-generated images than vice versa. The same holds for models fine-tuned using images from multiple GANs and DMs, respectively. We hypothesize that detectors trained solely on GAN-generated images focus mostly on GAN-specific (arguably more prominent) patterns, while detectors trained on DM-generated images focus on common patterns. This is supported by frequency (see Section 6) and feature space analyses (see Appendix B.3).

In addition, the results demonstrate that training a detector which is capable of correctly classifying both GAN- and DM-generated images is clearly possible (see rightmost column of Figure 1a). While this finding is promising, it is unclear whether a similar performance can be achieved in the model-agnostic setting addressed by Wang et al. (2020).

---

[3]In addition, we carry out the same evaluation for detectors trained from scratch, which leads to very similar results, as shown in Appendix B.2.

[4]`https://github.com/peterwang512/CNNDetection`

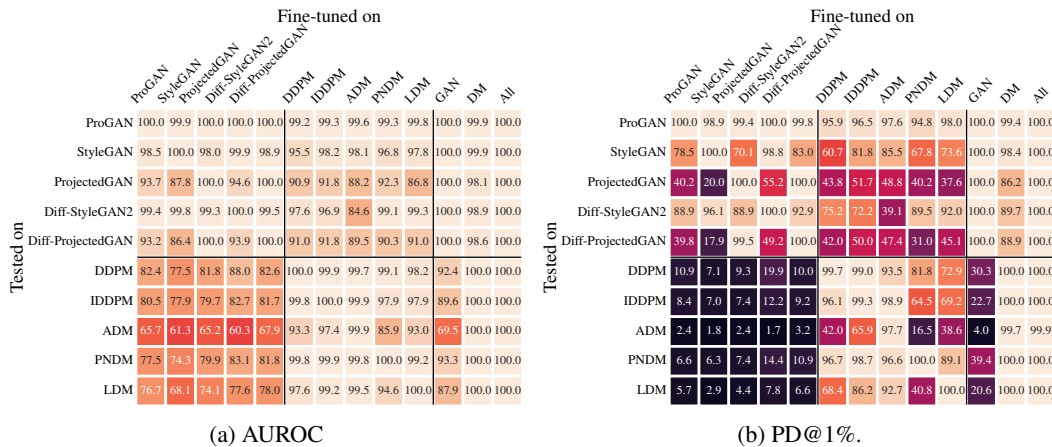

Figure 1: **Detection performance for fine-tuned detectors.** The columns *GAN*, *DM*, and *All* correspond to models fine-tuned on samples from all GANs, DMs, and both, respectively.

# 6 ANALYSIS OF DM-GENERATED IMAGES IN THE FREQUENCY DOMAIN

The visual quality of synthesized images has reached a level that makes them practically indistinguishable from real images for humans (Nightingale & Farid, 2022). However, in the case of GANs, the same does not hold when switching to the frequency domain (Zhang et al., 2019). In this section, we analyze the spectral properties of DM-generated images and compare them to those of GAN-generated images. We use three frequency transforms that have been used successfully in both traditional image forensics (Lyu, 2013) and deepfake detection: discrete Fourier transform (DFT), discrete cosine transform (DCT), and the reduced spectrum, which can be viewed as a 1D representation of the DFT. While DFT and DCT visualize frequency artifacts, the reduced spectrum can be used to identify spectrum discrepancies. The formal definitions of all transforms are provided in Appendix C.1. Furthermore, we evaluate the influence of different parameters during training and sampling of DMs and hypothesize on the origin of spectrum discrepancies.

## 6.1 SPECTRAL PROPERTIES OF DM- VS GAN-GENERATED IMAGES

Figure 2 depicts the average absolute DFT spectrum of 10k images from each GAN and DM in our dataset.[5] Before applying the DFT, images are transformed to grayscale and, following previous works (Marra et al., 2019; Wang et al., 2020), high-pass filtered by subtracting a median-filtered version of the image. For all GANs we observe significant artifacts, predominantly in the form of a regular grid. Taking the reduced spectra in Figure 3a into account we confirm the previously reported elevated high frequencies (see Section 2). For ProjectedGAN and Diff-ProjectedGAN, this characteristic is also visible. However, the spectral density does not exceed that of real images. Diff-StyleGAN2 is an exception as it shows a slight decrease towards the end of the spectrum.

In contrast, the DFT spectra of images generated by DMs (see Figure 2b), except LDM, are significantly more similar to the real spectrum with almost no visible artifacts. LDM, however, exhibits a thin grid across its spectrum, as well as the increased amount of high frequencies which is characteristic for GANs (see Figure 3b). As mentioned in Section 4, the architecture of LDM differs from the remaining DMs as the actual image is generated using an adversarially trained autoencoder, which could explain the discrepancies. This supports previous findings which suggest that the discriminator is responsible for spectrum deviations.

Although the reduced spectra of the remaining DMs do not exhibit the GAN-like rise towards the end of the spectrum, several DMs deviate stronger from the spectrum of real images than GANs. Especially for ADM, IDDPM, and DDPM, we observe an underestimation of the spectral density which becomes stronger towards higher frequencies (relative to the overall spectral density level). In Section 6.3 we discuss potential causes.

---

[5]We also report the DCT spectra of both GANs and DMs in Appendix C.2 which produce similar findings.

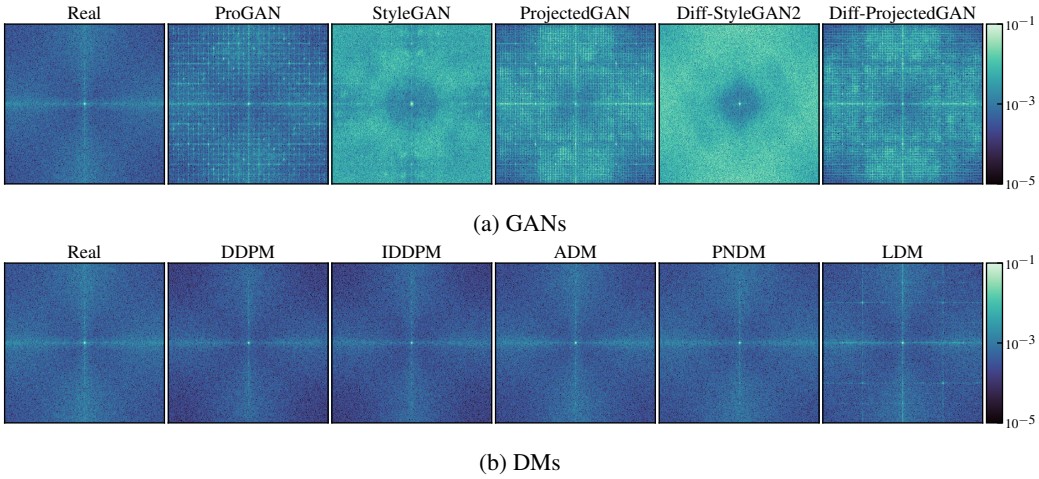

(a) GANs

(b) DMs

Figure 2: **Mean of DFT spectrum from real and generated images.** To increase visibility, the color bar is limited to $[10^{-5}, 10^{-1}]$, with values lying outside this interval being clipped.

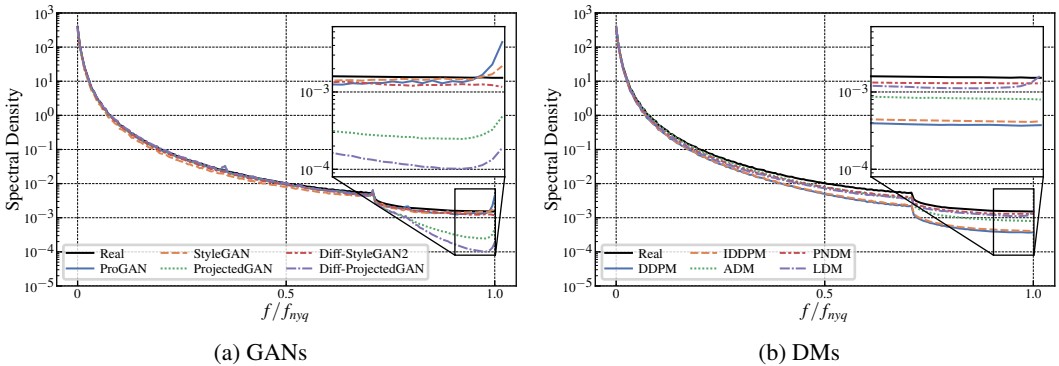

(a) GANs

(b) DMs

Figure 3: **Mean reduced spectrum from real and generated images.** The higher end of the spectrum where GAN-characteristic discrepancies occur is magnified.

## 6.2 ARE DM-GENERATED IMAGES EASIER TO DETECT IN THE FREQUENCY DOMAIN?

So far we showed that although DMs do not exhibit strong frequency artifacts, their spectrum deviates from that of real images. A natural question is therefore whether these discrepancies can be used to detect DM-generated images more effectively. Following Frank et al. (2020), we perform a simple logistic regression on each dataset with different transforms: Pixel (no transform), DFT (and taking the absolute value), and DCT. We use 20k samples for training, 2k for validation, and 20k for testing, each set equally split between real and fake images. To reduce the number of features, we transform all images to grayscale and take a center crop with dimensions 64x64. Additionally, all features are independently standardized to have zero mean and unit variance. We apply $L_2$ regularization and identify the optimal regularization weight by performing a grid search over the range $\left\{10^k \mid k \in \{-4, -3, \dots, 4\}\right\}$.

The accuracy of all GANs and DMs in our dataset is given in Table 3. We also report the results for log-scaled DFT and DCT coefficients, as this leads to significant improvements for some generators. For both GANs and DMs, using information from the frequency domain increases classification accuracy. On average, the performance gain of the best transform compared to no transform is 5.72% and 10.6%, respectively. Although the gain for DMs is more than double that for GANs, the overall maximum accuracy is significantly lower (90.9% for GANs and 63.1% for DMs on average). Therefore, we do not conclude that DMs exhibit stronger discriminative features in the frequency domain compared to GANs.

Table 3: **Accuracy of logistic regression on pixels and different transforms.** Next to each transform column we report the gain compared to the accuracy on pixels.

| | Pixel | DFT | | log(DFT) | | DCT | | log(DCT) | |
|---|---|---|---|---|---|---|---|---|---|
| ProGAN | 64.8 | 74.7 | +9.9 | 72.6 | +7.8 | 65.4 | +0.6 | **74.9** | +10.1 |
| StyleGAN | 91.1 | 87.4 | -3.7 | 86.2 | -4.9 | **92.6** | +1.5 | 86.4 | -4.7 |
| ProjectedGAN | 90.0 | 90.8 | +0.8 | 90.3 | +0.3 | 91.0 | +1.0 | **95.3** | +5.3 |
| Diff-StyleGAN2 | 92.4 | 80.3 | -12.0 | 80.5 | -11.9 | **93.8** | +1.4 | 87.6 | -4.7 |
| Diff-ProjectedGAN | 87.4 | 93.9 | +6.5 | 93.1 | +5.7 | 88.1 | +0.6 | **97.7** | +10.3 |
| DDPM | 51.7 | 64.2 | +12.6 | 64.2 | +12.5 | 52.4 | +0.7 | **64.3** | +12.6 |
| IDDPM | 51.6 | **62.1** | +10.6 | 61.7 | +10.1 | 51.7 | +0.1 | 61.7 | +10.1 |
| ADM | 50.1 | **54.7** | +4.6 | 52.3 | +2.3 | 50.1 | +0.0 | 53.8 | +3.7 |
| PNDM | 52.5 | 57.0 | +4.5 | 58.2 | +5.7 | 51.1 | -1.4 | **61.4** | +8.9 |
| LDM | 56.6 | 63.7 | +7.1 | 66.1 | +9.5 | 58.5 | +1.9 | **73.0** | +16.3 |

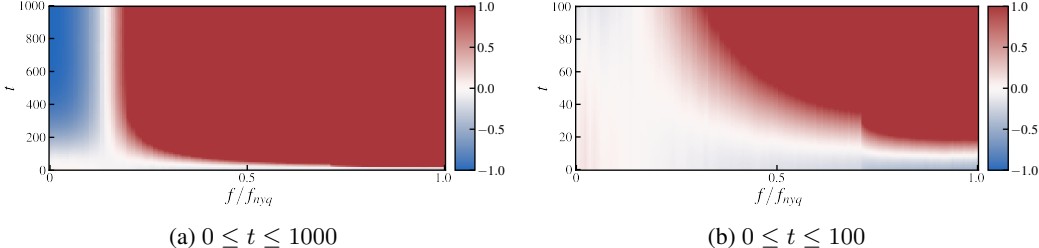

(a) $0 \leq t \leq 1000$                    (b) $0 \leq t \leq 100$

Figure 4: **Evolution of spectral density error throughout the denoising process.** The error is computed relatively to the spectrum of real images. We display the error for (a) all sampling steps and (b) a close-up of the last 100 steps. The colorbar is clipped at -1 and 1.

## 6.3 INVESTIGATING THE SOURCE OF SPECTRUM DISCREPANCIES

As explained in Section 2, several recent works attempted to identify the source of spectrum discrepancies in GAN-generated images. In this section, we make an effort to launch a similar line of work for DMs. For the experiments, we used code and model from ADM (Dhariwal & Nichol, 2021) pre-trained on LSUN Bedroom. Spectra are generated using 512 grayscale images.

**Spectrum Evolution During the Denoising Process** Our first analysis targets the denoising process. We generate samples at different time steps $t$ and compare their average reduced spectrum to that of 50k real images. The results are shown in Figure 4. We adopt the figure type from Schwarz et al. (2021) and depict the relative spectral density error $\tilde{S}_{\mathrm{err}} = \tilde{S}_{\mathrm{fake}}/\tilde{S}_{\mathrm{real}} - 1$, with the colorbar clipped at -1 and 1. At $t = 1000$, the image is pure Gaussian noise, which causes the strong spectrum deviations. Around $t = 300$, the error starts to decrease, but interestingly it appears to not continuously improve until $t = 0$. With regard to high frequencies, we observe an optimum at approximately $t = 10$ followed by an increasing underestimation. In Appendix C.3, we additionally provide the spectral density error between the denoising process and the diffusion process at the corresponding step, which further visualizes this behavior. However, at $t = 10$, the images are easier to detect and also visually noisier (see Appendix C.3 Figures 16 and 17). Thus, stopping the denoising process early is not an effective means to make synthetic images less detectable.

We hypothesize that the underestimation towards higher frequencies is due to a training objective that is suboptimal for capturing high frequency content. As noted by Nichol & Dhariwal (2021), the objective proposed by Ho et al. (2020), $L_{\mathrm{simple}} = \mathbb{E}_{t,\mathbf{x}_0,\epsilon}[\|\epsilon - \epsilon_\theta(\mathbf{x}_t, t)\|^2]$, does not lead to good log-likelihood values. These depend on high-frequency details synthesized at the challenging denoising tasks near $t = 0$ (Kingma et al., 2021). In contrast, the classical variational lower bound $L_{\mathrm{vlb}}$ forces the model to focus on the denoising steps towards $t = 0$. However, as $L_{\mathrm{vlb}}$ is extremely difficult to optimize (see e.g., Nichol & Dhariwal (2021)) the modified objective $L_{\mathrm{hybrid}} = L_{\mathrm{simple}} + \lambda L_{\mathrm{vlb}}$ (with $\lambda = 0.001$) is proposed by (Nichol & Dhariwal, 2021) and

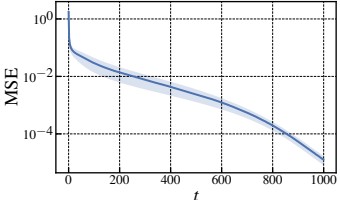

Figure 5: **Mean and standard deviation of the mean-squared error at different steps.**

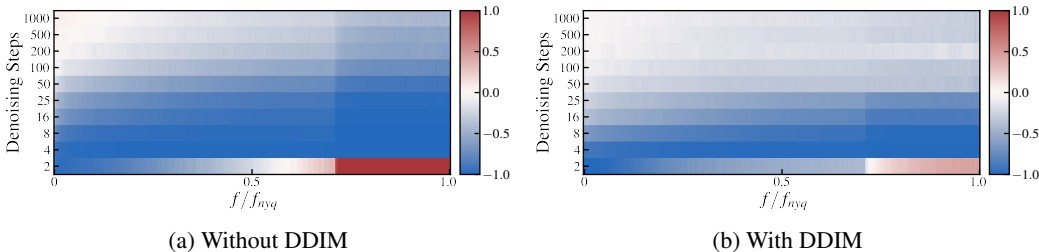

(a) Without DDIM                    (b) With DDIM

Figure 6: **Evolution of spectral density error for different numbers of denoising steps.** The error is computed relatively to the spectrum of real images. The colorbar is clipped at -1 and 1. Note that the y-axis is not scaled linearly.

also employed by (Dhariwal & Nichol, 2021). And indeed, when looking at the mean-squared error at different $t$ (see Figure 5), we observe a steep increase near $t = 0$, suggesting the model struggles with accurately removing the small perturbations which correspond to high frequencies. We conclude that the training objective used in state-of-the-art DMs enables impressive perceptual image quality (also regarding FID), but struggles to accurately model the high-frequency content of real images – which in turn can be exploited for detection. We elaborate in Appendix C.5.

**Effect of Number of Sampling Steps**  In a second step, we analyze how varying the number of sampling steps affects the spectral properties of generated images. While DMs are usually trained using $T = 1000$ steps, sampling speed can be increased by reducing the number of steps (Nichol & Dhariwal, 2021). For ADM on LSUN Bedroom, however, images were sampled using 1000 steps since this led to much better results (Dhariwal & Nichol, 2021). The authors also evaluate DDIM (Song et al., 2022a), an alternative sampling method which allows for sampling with fewer steps.

Using the previous experimental settings, we analyze the reduced spectra of images sampled using different numbers of steps (Figure 6). With only a few steps, the spectral density is too low across the entire spectrum, resulting in blurry images with little contrast (see Appendix C.4). Performing more steps leads to lower errors, with DDIM improving faster than normal sampling. This finding is coherent with previous results, as DDIM achieves better samples at fewer sampling steps (Song et al., 2022a). In Appendix C.4 we also analyze the detectability with different numbers of sampling steps. For both sampling methods, images become easier to detect with fewer sampling steps.

## 7  CONCLUSION

In this work, we make a much-needed first step towards the detection of DM-generated images. We find that existing detectors, which claim to be universal, fail to effectively recognize images synthesized by different DMs. This hints at fundamental differences between images generated by DMs and GANs, one of which we identify as different spectral properties. While DMs produce little to no frequency artifacts, there appears to be a systematic underestimation of high frequencies. Our hypothesis is that during training, too little weight is attached to these frequencies due to the choice of the training objective. However, this is an appropriate choice because correctly reproducing high frequencies is less important to the perceived quality of generated images than matching lower frequencies. Whether spectral discrepancies can be reduced (e.g., by adding spectral regularization (Durall et al., 2020)) without impairing image quality should be the subject of future work. Most importantly, more efforts are needed to develop novel detection methods, both universal and DM-specific. For tackling this task, a sufficiently large number of DM-generated images is necessary, which poses a challenge due to the slow generation speed compared to GANs. We therefore urge researchers presenting novel generation methods to publish sample images alongside pre-trained models, to enable the development and evaluation of detection methods without the need for immense computational resources. This is especially relevant due to the emergence of powerful text-to-image models based on DMs like DALL-E 2 (Ramesh et al., 2022), Imagen (Saharia et al., 2022), and Stable Diffusion (Rombach et al., 2022). We hope that our work can spark further research in this direction and emphasizes the fact that advances in realistic media synthesis should always be accompanied by considerations regarding deepfake detection methods.

ETHICS STATEMENT

While generative models have many beneficial applications, deepfakes pose a serious threat to society by undermining trust in media and digital communication (e.g., via fake profiles in social media that are used in misinformation campaigns). This work explores the challenges of deepfake detection for the novel promising class of diffusion models. We believe that the urge for reliable deepfake detection – ideally agnostic to the underlying generative model – is amplified by the ever easier use of generative models (e.g., via text-to-image synthesis). By highlighting the (current) lack of "universal" detectors, we strongly encourage the community to address this shortcoming, and our results can serve as a starting point for future work in this direction. While insights from such research could potentially be used to exacerbate or eventually elude detection, we strongly believe further insights in this area are much needed in order to sustain identifiability of deepfakes.

REPRODUCIBILITY STATEMENT

To ensure reproducibility, we provide instructions to recreate the dataset in Appendix A and, for the additional datasets, in Appendix D. All datasets considered in this work are publicly available. We train/fine-tune the detector proposed by Wang et al. (2020) using the settings provided by the authors and describe training and validation data in the text (Section 5.2). All tables and figures can be reproduced using the source code provided in the supplementary materials.

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

## A  DETAILS ON THE DATASET

**LSUN Bedroom (Yu et al., 2016)**  We download and extract the lmbd database files using the official repository[6]. The images are center-cropped to 256x256 pixels.

**ProGAN (Karras et al., 2018)**  We download the first 10k samples from the non-curated collection provided by the authors.[7]

**StyleGAN (Karras et al., 2019)**  We download the first 10k samples generated with $\psi = 0.5$ from the non-curated collection provided by the authors.[8]

**ProjectedGAN (Sauer et al., 2021)**  We sample 10k images using code and pre-trained models provided by the authors using the default configuration (`--trunc=1.0`).[9]

**Diff-StyleGAN2 (Wang et al., 2022) and Diff-ProjectedGAN (Wang et al., 2022)**  We sample 10k images using code and pre-trained models provided by the authors using the default configuration.[10]

**DDPM (Ho et al., 2020), IDDPM (Nichol & Dhariwal, 2021), and ADM (Dhariwal & Nichol, 2021)**  We download the samples provided by the authors of ADM[11] and extract the first 10k samples for each generator. For ADM on LSUN, we select the models trained with dropout.

**PNDM (Liu et al., 2022)**  We sample 10k images using code and pre-trained model provided by the authors.[12] We specify `--method F-PNDM` and `--sample_speed 20` for LSUN Bedroom and `--sample_speed 10` for LSUN Church, as these are the settings leading to the lowest FID according to Tables 5 and 6 in the original publication.

**LDM (Rombach et al., 2022)**  We sample 10k images using code and pre-trained models provided by the authors using settings from the corresponding table in the repository.[13] For LSUN Church there is an inconsistency between the repository and the paper, we choose 200 DDIM steps (`-c 200`) as reported in the paper.

---

[6]`https://github.com/fyu/lsun`
[7]`https://github.com/tkarras/progressive_growing_of_gans`
[8]`https://github.com/NVlabs/stylegan`
[9]`https://github.com/autonomousvision/projected_gan`
[10]`https://github.com/Zhendong-Wang/Diffusion-GAN`
[11]`https://github.com/openai/guided-diffusion`
[12]`https://github.com/luping-liu/PNDM`
[13]`https://github.com/CompVis/latent-diffusion`

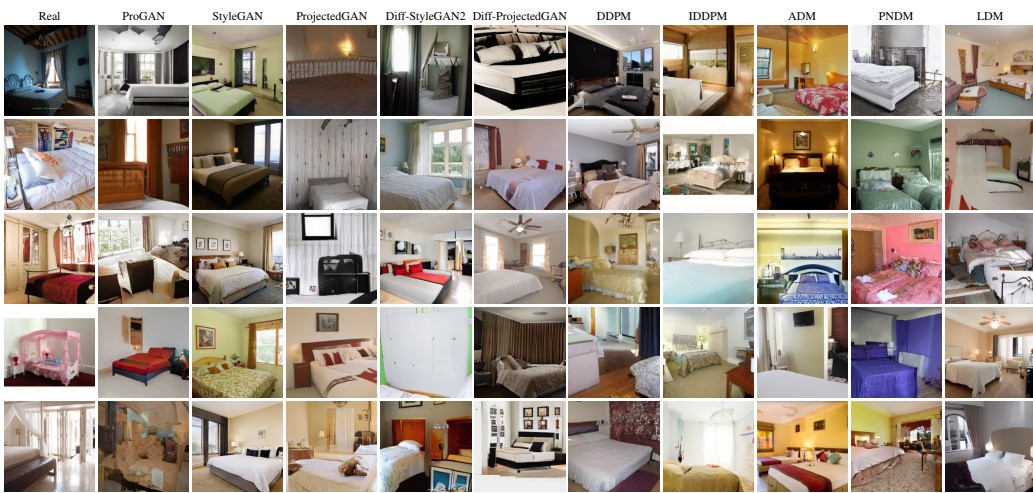

Figure 7: **Non-curated examples for real LSUN Bedroom, GAN-generated, and DM-generated images.**

# B DETECTION

## B.1 DESCRIPTIONS OF DETECTORS

**Wang et al. (2020)** In this influential work, the authors demonstrate that a standard deep convolutional neural network (CNN) trained on data from a single generator performs surprisingly well on unseen images. They train a ResNet-50 (He et al., 2016) on 720k images from 20 LSUN (Yu et al., 2016) categories (not including Bedroom and Church), equally divided into real images and images generated by ProGAN (Karras et al., 2018). The trained binary classifier is able to distinguish real from generated images from a variety of generative models and datasets. The authors further show that extensive data augmentation in the form of random flipping, blurring, and JPEG compression generally improves generalization. Moreover, the authors provide two pre-trained model configurations, Blur+JPEG (0.1) and Blur+JPEG (0.5), where the value in parentheses denotes the probability of blurring and JPEG compression, respectively. They achieve an average precision of 92.6% and 90.8%, respectively. The work suggests that CNN-generated images contain common artifacts (or fingerprints) which make them distinguishable from real images.

**Gragnaniello et al. (2021)** Building upon the architecture and dataset from Wang et al. (2020), the authors of this work experiment with different variations to further improve the detection performance in real-world scenarios. The most promising variant, no-down, removes downsampling from the first layer, increasing the average accuracy from 80.71% to 94.42%, at the cost of more trainable parameters. They also train a model with the same architecture on images generated by StyleGAN2 (Karras et al., 2020) instead of ProGAN, which further improves accuracy to 98.48%.

**Mandelli et al. (2022a)** Unlike the other two methods, this work uses an ensemble of five orthogonal CNNs to detect fake images not seen during training. All CNNs are based on the EfficientNet-B4 model (Tan & Le, 2019) but are trained on different datasets. Dataset orthogonality refers to images having different content, different processing (e.g., JPEG compression), or being generated by different GANs. They argue that by having different datasets, each CNN learns to detect different characteristics of real and generated images, improving the overall performance and generalization ability. For each CNN, a score is computed for $\approx 200$ random patches extracted from the image. These scores are combined using a novel patch aggregation strategy which assumes that an image is fake if at least one patch is classified as being fake. Finally, the output score is computed by averaging the individual scores of all five CNNs. It should be noted that, besides several GANs, the training dataset also includes samples generated by Song et al. (2022b), a score-based model.

## B.2 DETECTION RESULTS FOR CLASSIFIERS TRAINED FROM SCRATCH

We repeat the experiment described in Section 5.2 but train each classifier from scratch instead of fine-tuning it, the results are shown in Figure 8. Note that here "from scratch" means that we start from a ResNet-50 model pre-trained on ImageNet, which is exactly how the authors trained their models (Wang et al., 2020). For most constellations of training and test data, the results are similar compared to fine-tuning, with a tendency towards worse scores for the models trained from scratch.

## B.3 FEATURE SPACE ANALYSIS

The experiments in Section 5.1 demonstrate that current detection methods struggle to reliably identify DM-generated images while detecting GAN-generated images with near-optimal accuracy. We thus hypothesize that GAN-generated images exhibit features that are considerably distinct from those of DM-generated images.

To further investigate this hypothesis we examine the discrepancy of real and fake images in the feature spaces of the detection methods under consideration. More precisely, we employ Maximum Mean Discrepancy (MMD; Gretton et al., 2012) to estimate the distance between the distributions of real and generated images in feature space. We extract the 2048-dimensional features before the last fully-connected layer of the detectors by Wang et al. (2020) and Gragnaniello et al. (2021). We employ a Gaussian kernel with $\sigma$ as the median distance between instances in the combined sample as suggested in Gretton et al. (2012) and calculate the MMD between the representations of 10k generated images and 10k real images.

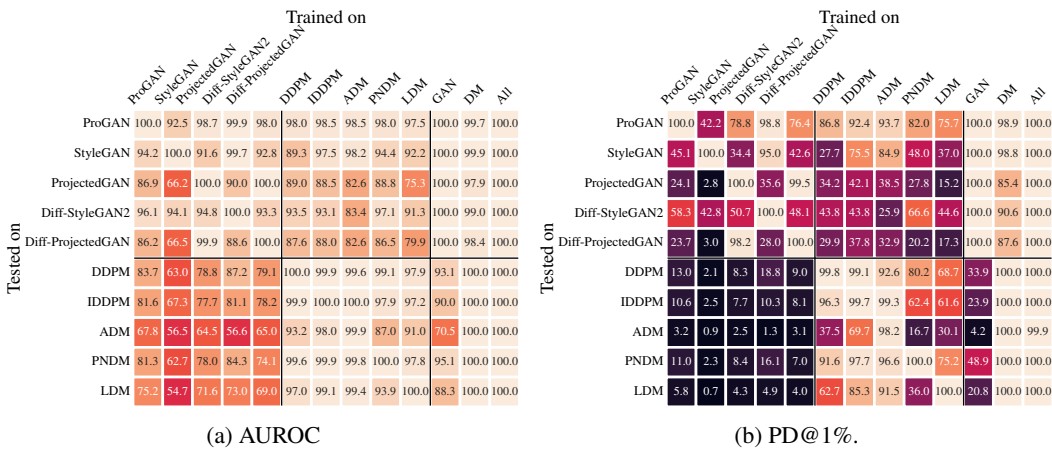

(a) AUROC

(b) PD@1%.

Figure 8: **Detection performance for detectors trained from scratch.**

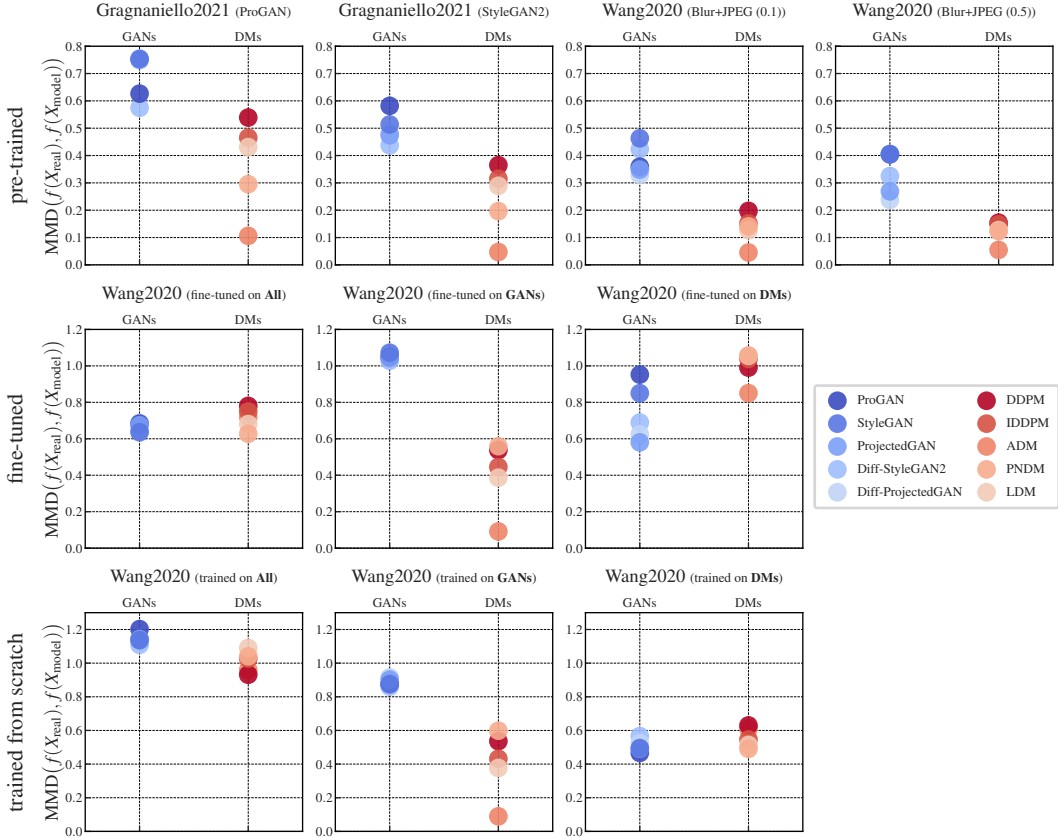

Figure 9: **MMD between feature representations of real and generated images on LSUN Bedroom for different detectors (pre-trained, fine-tuned and trained from scratch)**. The features $f(\cdot)$ correspond to the representation prior to the last fully-connected layer of the respective detection method. For further information on the detectors see Section 5.2 (fine-tuned) and Appendix B.2 (trained from scratch).

We consider three sets of detection methods, namely pre-trained, fine-tuned and trained from scratch.

Starting with pre-trained detection methods (top row in Figure 9), we observe that the MMD between representations of DM-generated images and those of real images are considerably lower in comparison to GAN-generated images. With the results from Table 2 in mind, we can conclude that the pre-trained detectors extract features that allow to reliably separate GAN-generated images from real images, while these features are less informative to discriminate DM-generated images. This findings support the experiments conducted in Section 5.1.

We repeat the above experiment for fine-tuned and re-trained detectors from Wang et al. (2020) (middle row and bottom row in Figure 9). We consider the version Blur+JPEG (0.5) and fine-tune/train on all GAN-generated images, DM-generated images and both (as described in Section 5.2 and Appendix B.2). The results of fine-tuned and re-trained detectors are qualitatively similar: When training/fine-tuning on generated images from both, DMs and GANs, the MMDs in feature space are roughly on par for both model sets. This strengthens the finding that reliable detection is feasible provided the generative model class is known. When fine-tuning/training solely on images from one model class we observe an imbalance: Detectors fine-tuned or trained on images from DMs are able to achieve relatively higher MMDs for GAN-generated images than vice versa. This imbalance is already evident in the experiments in Section 5.2.

The findings give rise to the following hypothesis: GAN- and DM-generated images share some common patterns. However, detectors trained solely on GAN-generated images focus mostly on GAN-specific (arguably more prominent) patterns, while detectors trained on DM-generated images focus on common patterns. The hypothesis is corroborated by the t-SNE visualizations (van der Maaten & Hinton, 2008) provided in Figure 10.[14] The detectors trained on DM-generated images evidently learn features/representations which are common to both GANs and DMs (right column), while detectors which saw GAN-generated images during training map GAN- and DM-generated images to distinct representations (center columns). Moreover, detectors trained solely on GAN-generated images lead to representations of real and DM-generated images which are not clearly separated, which fits the reduced performance.

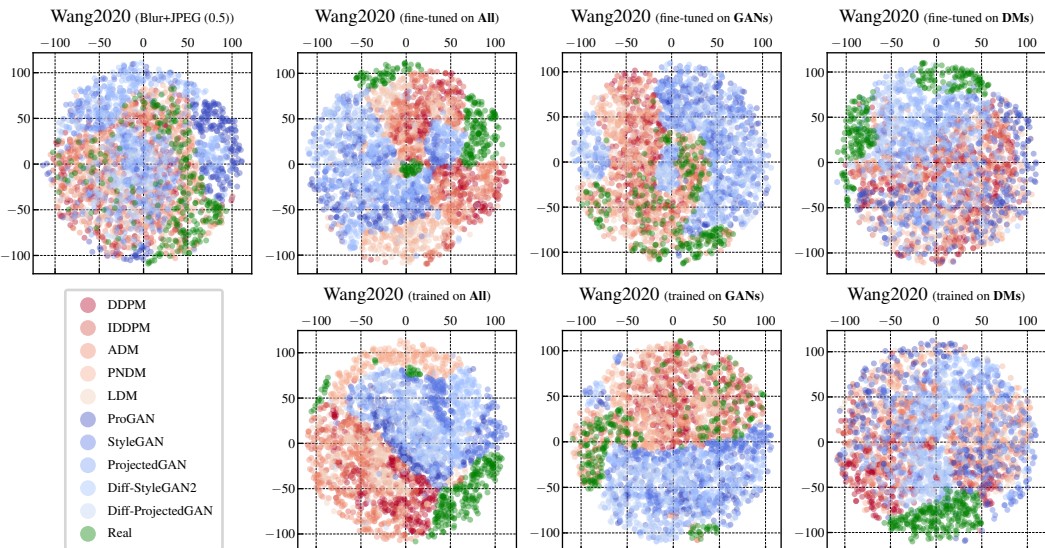

Figure 10: **t-SNE visualization of representations of real and generated images on LSUN Bedroom for different detectors (pre-trained, fine-tuned and trained from scratch) in two dimensions**. The representations/features $f(\cdot)$ correspond to the representation prior to the last fully-connected layer of the respective detection method and are of dimension $2048$. For further information on the detectors see Section 5.2 (fine-tuned) and Appendix B.2 (trained from scratch).

---

[14]We used the scikit-learn implementation of t-SNE with the default settings, details are given at `https://scikit-learn.org/stable/modules/generated/sklearn.manifold.TSNE.html`

Table 4: **Effect of image perturbations on detection performance.** We use the pre-trained detector by Gragnaniello et al. (2021) trained on ProGAN images and compute metrics from 10k samples.

| AUROC / Pd@5% / Pd@1% | Clean | Blur | Crop | JPEG | Noise |
|---|---|---|---|---|---|
| ProGAN | 100.0 / 100.0 / 100.0 | 100.0 / 100.0 / 100.0 | 100.0 / 100.0 / 100.0 | 100.0 / 100.0 / 100.0 | 99.2 / 95.8 / 82.5 |
| StyleGAN | 100.0 / 100.0 / 100.0 | 99.7 / 98.2 / 94.3 | 100.0 / 100.0 / 99.9 | 100.0 / 100.0 / 99.9 | 94.3 / 67.0 / 40.9 |
| ProjectedGAN | 100.0 / 99.9 / 99.3 | 99.2 / 96.3 / 87.6 | 99.9 / 99.7 / 98.0 | 100.0 / 100.0 / 99.6 | 96.6 / 79.9 / 52.8 |
| Diff-StyleGAN2 | 100.0 / 100.0 / 100.0 | 100.0 / 100.0 / 100.0 | 100.0 / 100.0 / 100.0 | 100.0 / 100.0 / 100.0 | 96.7 / 79.5 / 56.3 |
| Diff-ProjectedGAN | 99.9 / 99.9 / 99.2 | 99.0 / 94.9 / 84.1 | 99.9 / 99.7 / 97.0 | 100.0 / 100.0 / 99.4 | 96.1 / 77.0 / 51.0 |
| DDPM | 96.5 / 79.4 / 39.1 | 78.5 / 23.7 / 8.1 | 95.5 / 73.2 / 35.2 | 96.0 / 76.1 / 37.3 | 85.1 / 34.0 / 11.2 |
| IDDPM | 94.3 / 64.8 / 25.7 | 75.3 / 18.5 / 5.5 | 93.5 / 62.0 / 24.6 | 93.5 / 60.7 / 23.4 | 80.9 / 25.0 / 7.0 |
| ADM | 77.8 / 20.7 / 5.2 | 66.0 / 10.1 / 3.0 | 78.3 / 21.4 / 5.5 | 76.8 / 18.7 / 4.4 | 64.8 / 8.6 / 1.7 |
| PNDM | 91.6 / 52.0 / 16.6 | 86.7 / 38.8 / 14.3 | 91.1 / 51.9 / 19.7 | 92.5 / 56.6 / 20.6 | 81.2 / 27.4 / 9.3 |
| LDM | 96.7 / 79.9 / 42.1 | 87.1 / 42.8 / 17.1 | 96.8 / 81.8 / 48.9 | 96.3 / 77.3 / 40.0 | 82.2 / 28.6 / 8.8 |

Table 5: **Effect of image perturbations on detection performance.** We use the detector by Wang et al. (2020) fine-tuned on all DM-generated images and compute metrics from 10k samples.

| AUROC / Pd@5% / Pd@1% | Clean | Blur | Crop | JPEG | Noise |
|---|---|---|---|---|---|
| DDPM | 100.0 / 100.0 / 100.0 | 100.0 / 100.0 / 99.9 | 100.0 / 100.0 / 99.9 | 100.0 / 100.0 / 99.7 | 69.8 / 22.7 / 11.3 |
| IDDPM | 100.0 / 100.0 / 100.0 | 100.0 / 100.0 / 100.0 | 100.0 / 100.0 / 100.0 | 100.0 / 100.0 / 99.7 | 70.2 / 22.5 / 11.1 |
| ADM | 100.0 / 100.0 / 99.7 | 99.9 / 99.9 / 99.0 | 100.0 / 100.0 / 99.5 | 99.6 / 98.2 / 90.8 | 70.3 / 23.2 / 10.8 |
| PNDM | 100.0 / 100.0 / 100.0 | 100.0 / 99.9 / 99.8 | 100.0 / 100.0 / 100.0 | 100.0 / 100.0 / 99.7 | 74.8 / 29.2 / 14.6 |
| LDM | 100.0 / 100.0 / 100.0 | 100.0 / 100.0 / 100.0 | 100.0 / 100.0 / 100.0 | 100.0 / 100.0 / 99.6 | 71.9 / 25.7 / 12.9 |

## B.4 EFFECT OF IMAGE PERTURBATIONS

In most real-world scenarios, like uploading to social media, images are processed, which is why several previous works consider common image perturbations when evaluating detectors (Wang et al., 2020; Hulzebosch et al., 2020; Liu et al., 2020; Frank et al., 2020). We follow the protocol of Frank et al. (2020) and apply blurring using a Gaussian filter (kernel size sampled from $\{3, 5, 7, 9\}$, cropping with subsequent upsampling (crop factor sampled from $U(5, 20)$), JPEG compression (quality factor sampled from $U(10, 75)$), and Gaussian noising (variance sampled from $U(5, 20)$). Unlike Frank et al. (2020), we apply each perturbation with a probability of 100% to study its effect on the detection performance.

Table 4 shows the results for the best-performing detector by Gragnaniello et al. (2021) trained on ProGAN images. Note that this detection method employs training augmentation using blurring and JPEG compression. We copy the results on "clean" images from Table 2 as a reference. For GAN-generated images, blurring, cropping, and compression have only a very small effect on the detection performance, with the latter even improving the results. We attribute this to the fact that the detector uses data augmentation during training. While cropping and compression cause an average decrease in AUROC of 0.34% and 0.36%, respectively, blurring leads to a significant performance degradation of 12.66%. The addition of noise deteriorates both GAN- and DM-generated images, with AUROC decreasing by 3.4% and 12.54% on average, respectively. Overall, we observe that perturbations have a stronger effect on DM-generated images compared to GAN-generated images. An explanation could be that the pre-trained detector performs significantly worse on DM-generated images, even without any perturbations. We repeat the experiment using the detector by Wang et al. (2020) fine-tuned on images from all DMs (see Section 5.2). As the results in Table 5 show, the effect of blurring, cropping, and JPEG compression is almost negligible. Adding noise, however, causes a significant performance drop. This could to be related to the fact that the image generation process of DMs involves noise. However, another potential explanation is that the detector is shown blurred and compressed images during training but not those with added noise.

## B.5 FAKENESS RATING

We evaluate whether DM-generated images exhibit some visual cues used by the detector to distinguish them from real images. Inspired by Wang et al. (2020), we rank all images by the model's predictions (higher value means "more fake") and show examples from different percentiles. We consider two detectors, the best-performing pre-trained detector from Gragnaniello et al. (2021) trained on ProGAN (Table 11) and the detector from Wang et al. (2020) fine-tuned on all DM-generated images (Table 12). Details on the fine-tuning process are given in Section 5.2. Using the

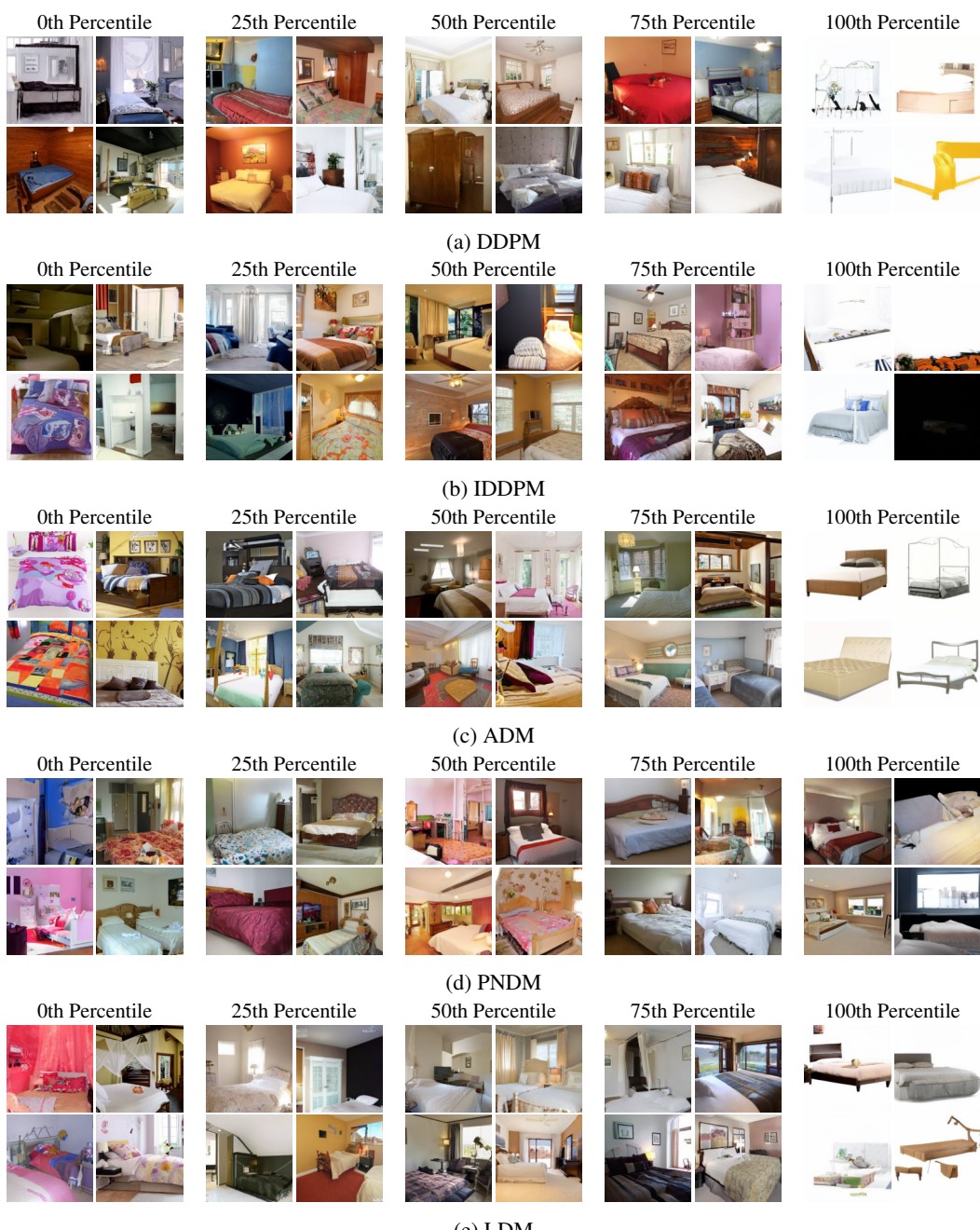

Figure 11: **"Fakeness" rating based of LSUN Bedroom on predictions from detector by Grag-naniello et al. (2021) trained on ProGAN images.** Images are ranked by the model's output, i.e., images in the 100th percentile are considered most fake.

detector from Gragnaniello et al. (2021), we make the observation that for most DMs, images which the model assigns a high "fakeness" score contain many pixels which are purely white or black. On the other hand, images considered less fake appear to be more colorful. We were able to reproduce this behavior for other datasets in Figure 13. However, this finding does not hold for the ranking provided by the fine-tuned detector (see Figure 12), which *should* provide more accurate results as its detection performance is significantly higher. Overall, we agree with Wang et al. (2020) that there is no strong correlation between the model's predictions and visual quality.

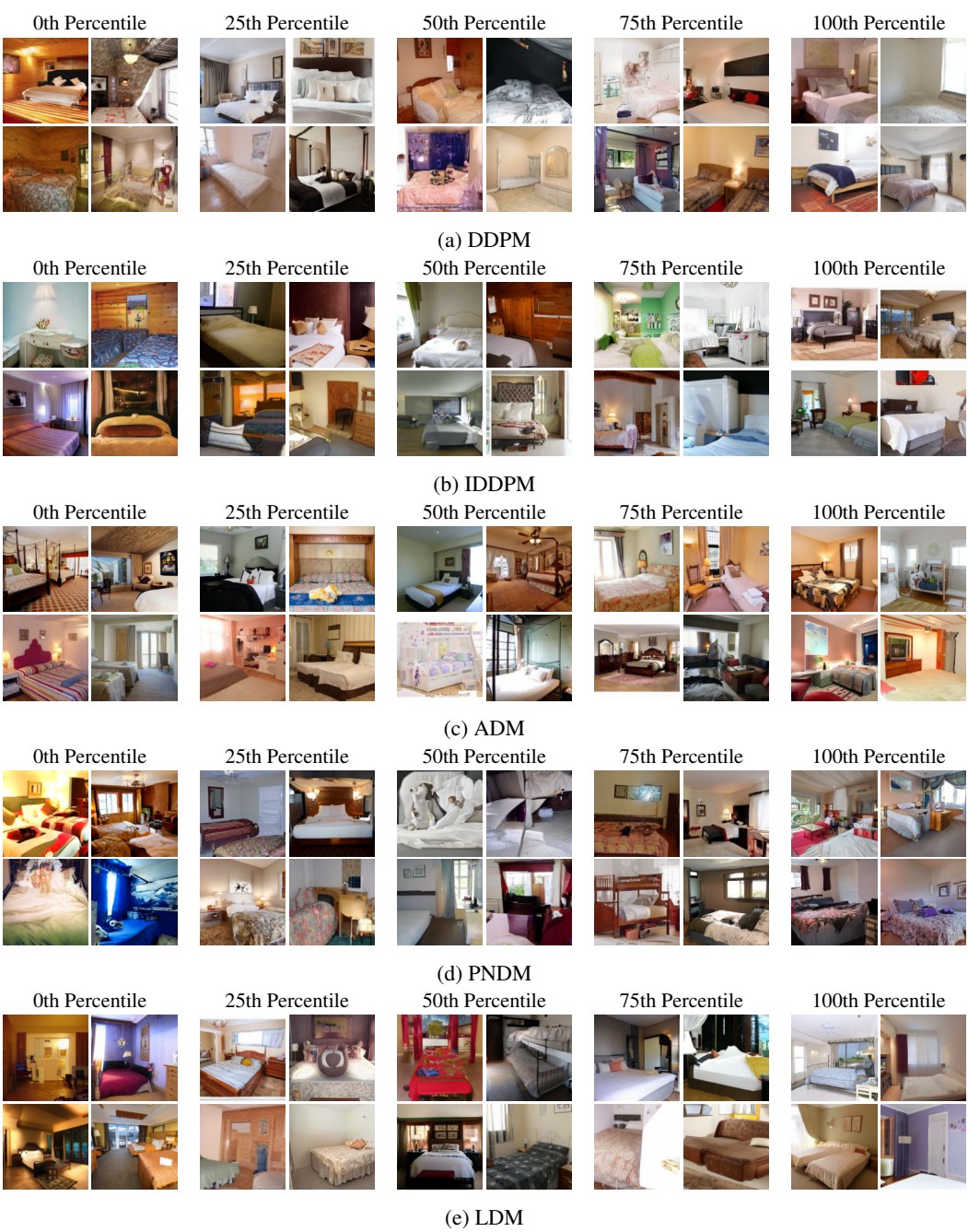

Figure 12: **"Fakeness" rating of LSUN Bedroom based on predictions from detector by Wang et al. (2020) fine-tuned on images from all DMs.** Images are ranked by the model's output, i.e., images in the 100th percentile are considered most fake.

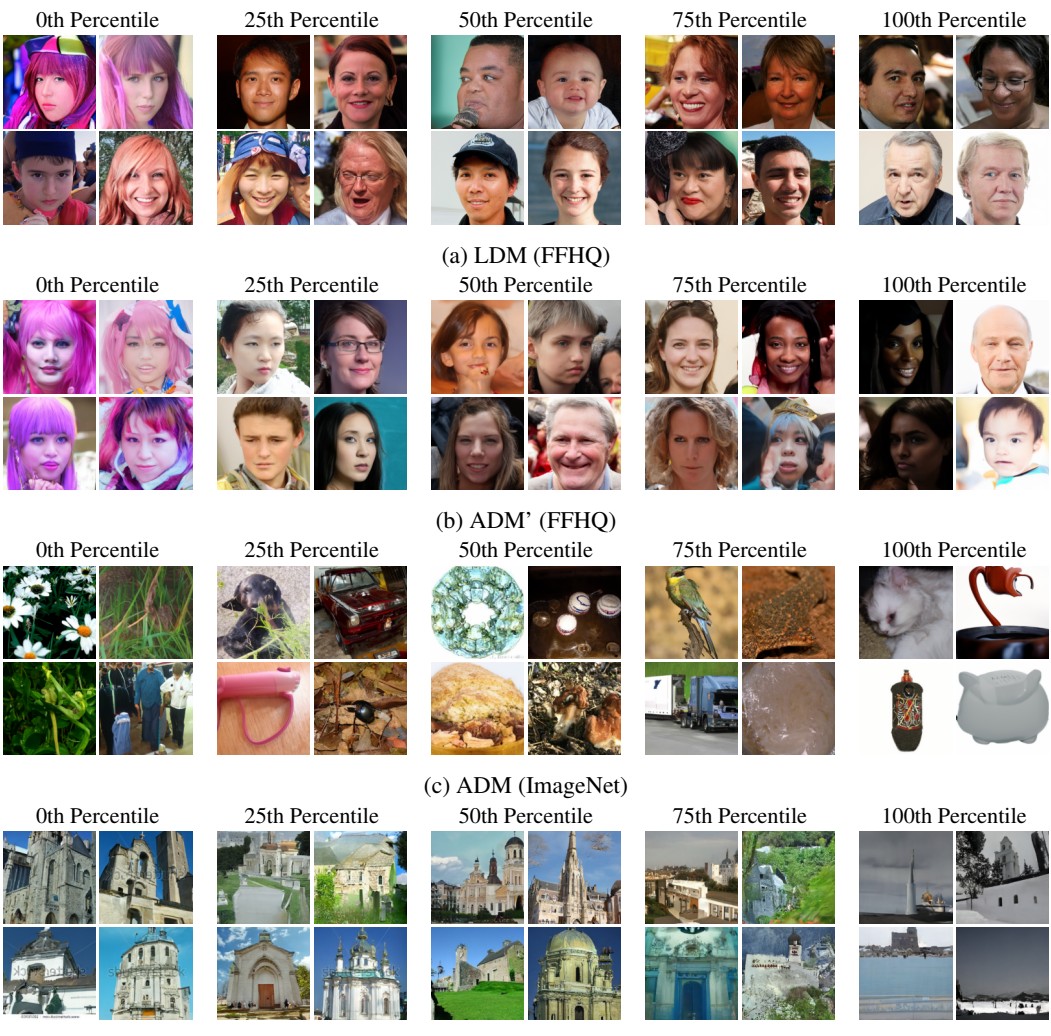

(a) LDM (FFHQ)

(b) ADM' (FFHQ)

(c) ADM (ImageNet)

(d) PNDM (LSUN Church)

Figure 13: **"Fakeness" rating of DM-generated images based on predictions from detector by Gragnaniello et al. (2021) trained on ProGAN images.** Images are ranked by the model's output, i.e., images in the 100th percentile are considered most fake.

## C  Frequency Analysis

### C.1  Frequency Transforms

**Discrete Fourier Transform (DFT)**   The DFT maps a discrete signal to the frequency domain by expressing it as a sum of periodic basis functions. Given a grayscale image $I$ with height $H$ and width $W$, the two-dimensional DFT (with normalization term omitted) is defined as

$$I_{\text{DFT}}[k, l] = \sum_{x=0}^{H-1} \sum_{y=0}^{W-1} I[x, y] \ \exp^{-2\pi i \frac{x \cdot k}{H}} \ \exp^{-2\pi i \frac{y \cdot l}{W}}, \tag{4}$$

with $k = 0, \dots, H - 1$ and $l = 0, \dots, W - 1$. For visualization, the zero-frequency component is shifted to the center of the spectrum. Therefore, coefficients towards the edges of the spectrum correspond to higher frequencies.

**Discrete Cosine Transform (DCT)**   The DCT is closely related to the DFT, however it uses real-valued cosine functions as basis functions. It is used in the JPEG compression standard due to its high degree of energy compaction (Wallace, 1991), which ensures that a large portion of a signal's energy can be represented using only a few DCT coefficients. The type-II DCT, which the term DCT usually refers to, is given as

$$I_{\text{DCT}}[k, l] = \sum_{x=0}^{H-1} \sum_{y=0}^{W-1} I[x, y] \cos\left[ \frac{\pi}{H} \left( x + \frac{1}{2} \right) k_x \right] \cos\left[ \frac{\pi}{W} \left( y + \frac{1}{2} \right) k_y \right], \tag{5}$$

again omitting the normalization factor. In the resulting spectrum, the low frequencies are located in the upper left corner, with frequencies increasing along both spatial dimensions.

**Reduced Spectrum**   While the DFT provides a useful visual representation of an image's spectrum, it is less suitable for comparing different spectra quantitatively. Therefore, previous works use the reduced spectrum[15], a one-dimensional representation of the Fourier spectrum (Durall et al., 2020; Dzanic et al., 2020; Schwarz et al., 2021; Chandrasegaran et al., 2021). It is obtained by azimuthally averaging over the spectrum in normalized polar coordinates $r \in [0, 1]$, $\theta \in [0, 2\pi)$ according to

$$\tilde{S}(r) = \frac{1}{2\pi} \int_0^{2\pi} S(r, \theta) d\theta \quad \text{with} \quad r = \sqrt{\frac{k^2 + l^2}{\frac{1}{4}(H^2 + W^2)}} \quad \text{and} \quad \theta = \text{atan2}(k, l), \tag{6}$$

with $S[k, l] = |I_{DFT}[k, l]|^2$ being the squared magnitudes of the Fourier coefficients. The maximum frequency is given by the Nyquist frequency $f_{\text{nyq}} = \sqrt{k^2 + l^2} = H/\sqrt{2}$ for a square image with $H = W$.

### C.2  DCT Spectra

Figure 14 depicts the DCT spectra of both GANs and DMs. Similar to DFT, images generated by DMs exhibit fewer artifacts, with the exception of LDM.

### C.3  Spectrum Evolution During the Denoising Process

Analogous to Figure 4, we show the spectrum error evolution during the denoising process in Figure 15. Here, however, the error is computed relative to the spectrum of noised images at the corresponding step during the diffusion process. Similar to the denoising process, the spectra of the diffusion process are averaged over 512 samples. While the relative error is close to zero for a long time during the denoising process, the model fails to accurately reproduce higher frequencies towards $t = 0$. More precisely, too much high-frequency components are removed, which explains why the sweet spot does not seem to be at $t = 0$ but around $t = 10$ (see Figure 4).

---

[15]The definition of the reduced spectrum slightly differs between different existing works, we decide to follow that of Schwarz et al. (2021).

One could think that, by stopping the denoising process early, DM-generated images might be harder to detect. To test this hypothesis, we perform a logistic regression at every $0 \leq t \leq 100$ to distinguish real from increasingly denoised generated images. The model is trained using 512 real and 512 generated samples, from which 20% are used for testing. We select the optimal regularization weight by performing a 5-fold cross-validation over $\{10^k \mid k \in \{-4, -3, \ldots, 4\}\}$. Similar to Section 6.2, we compare the performance on pixels and different transforms. The results in Figure 16 do not indicate that around $t = 10$ fake images are less detectable. In Figure 17 it becomes apparent that at this $t$ the images are noticeably noisier, which probably explains the increased accuracy.

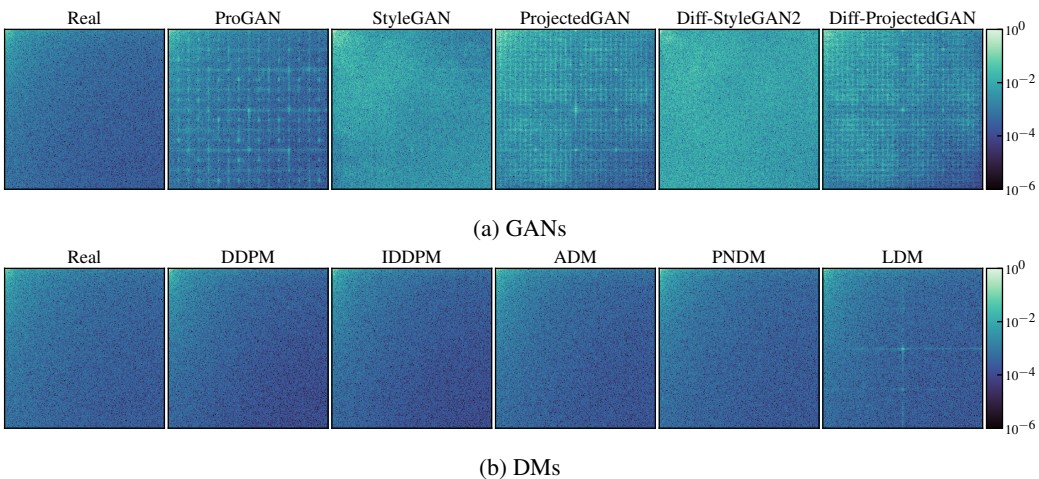

Figure 14: **Mean of DCT spectrum from real and generated images.** To increase visibility, the color bar is limited to $[10^{-6}, 10^0]$, with values lying outside this interval being clipped.

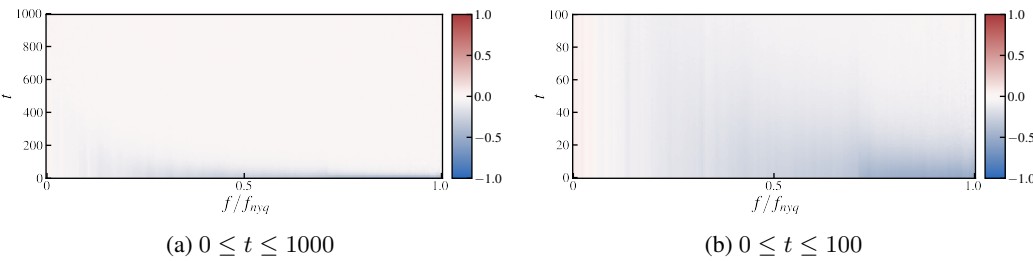

Figure 15: **Evolution of spectral density error throughout the denoising process.** The error is computed relatively to the corresponding step in the diffusion process. The colorbar is clipped at -1 and 1. If the model was able to perfectly denoise, the difference would be 0 at all $t$.

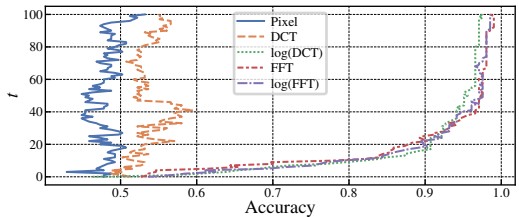

Figure 16: **Accuracy of logistic regression during the denoising process.** Note that only the last 100 steps of the denoising process are depicted.

## C.4    EFFECT OF NUMBER OF SAMPLING STEPS

We repeat the logistic regression experiment from the previous section for samples generated with different numbers of timesteps, the results are shown in Figure 18. We also provide example images in Figure 19. As expected, images become harder to detect with more sampling steps and therefore higher image quality. For DDIM, the accuracy decreases more quickly, which is likely because it was found to generate images of higher quality with fewer sampling steps (Song et al., 2022a).

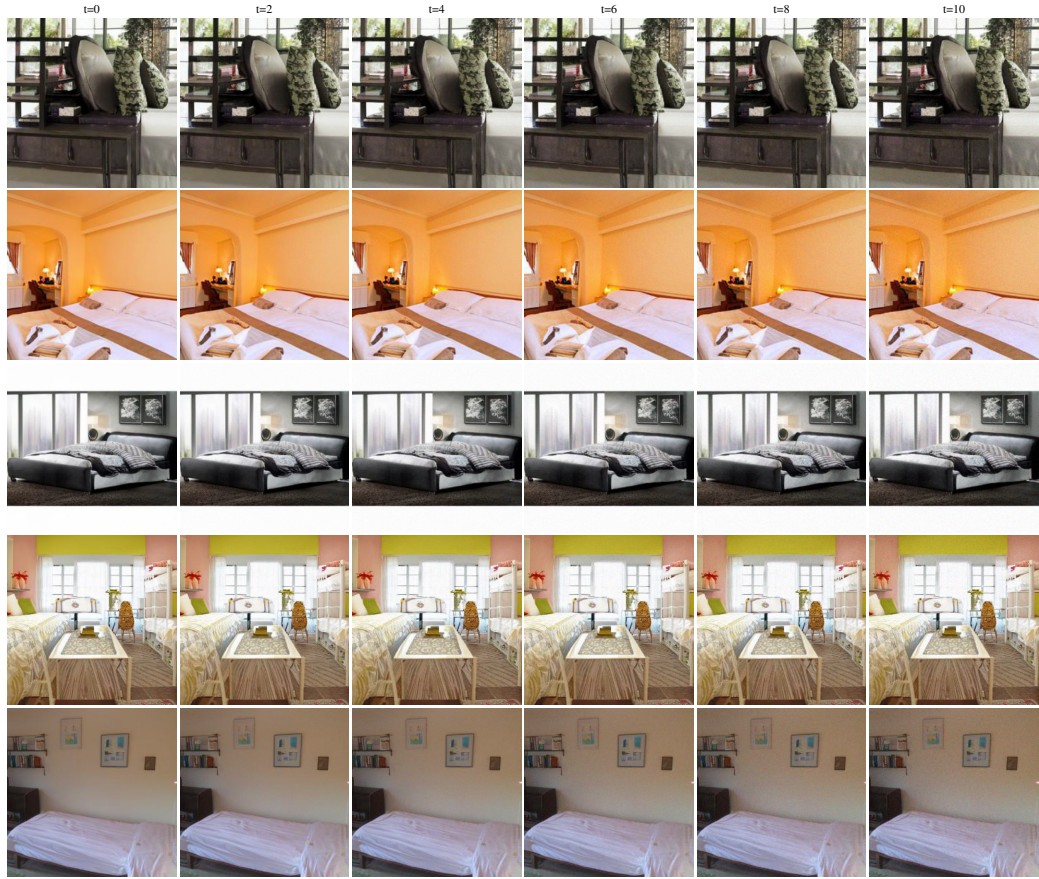

Figure 17: **Example images generated by ADM at different t.** When zoomed in, the high-frequency noise towards higher $t$ becomes apparent.

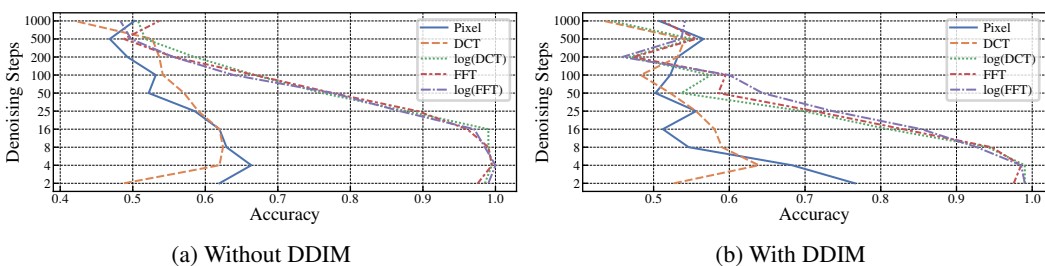

(a) Without DDIM                    (b) With DDIM

Figure 18: **Accuracy of logistic regression for different numbers of sampling steps.** Note that the y-axis is not scaled linearly.

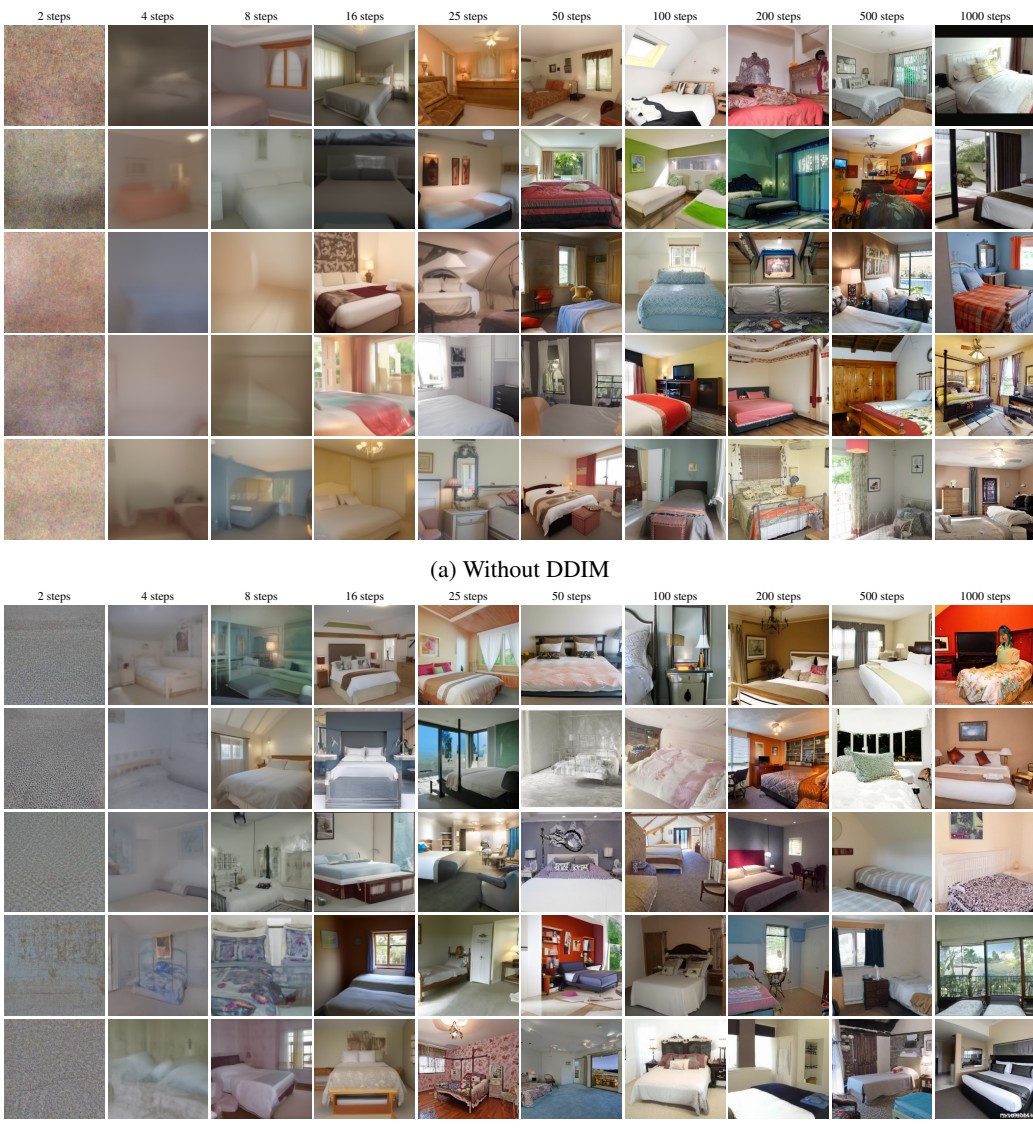

Figure 19: **Example images generated by ADM with different numbers of sampling steps.**

## C.5 Potential Source of the Spectrum Discrepancies

Our analysis in the frequency domain (see Figures 3 and 23) suggests that current state-of-the-art DMs do not match the high-frequency content well. To further analyze these findings, we build on insights from the denoising autoencoder (DAE) literature. Roughly speaking, the task DAEs face is conceptually similar to the task of the noise predictor $\epsilon_\theta$ in DMs at a single time step: denoising a disturbed input at a fixed noise level. Note that while we believe that DMs and DAEs can be conceptually related, the concepts are distinct in several ways: DMs use parameter sharing to perform noise prediction at multiple noise levels to set up the generation as an iterative process. On the other hand, DAEs make use of a latent space to learn suitable representations for reconstruction, classically at a fixed noise level. Nevertheless, we are convinced that it may be useful to take these insights into account.

A handy observation from DAEs relates the level of corruption to the learned feature scales: Denoising an image at small noise levels requires to accurately model fine granular details of the image, while coarse/large-scale features can still be recovered at high noise levels (see e.g., Vincent et al.

(2010); Geras & Sutton (2015; 2016)). Transferring this insight to DMs, we observe that the training objective guides the reconstruction performance across the different noise levels.

Recall that the objective of many prominent DMs can be stated as a weighted sum of mean squared error terms

$$L(\theta) = \sum_{t=0}^{T} w(t) \mathbb{E}_{t,\mathbf{x}_0,\epsilon}[\|\epsilon - \epsilon_\theta(\mathbf{x}_t, t)\|^2] \tag{7}$$

usually with $T = 1000$. The theoretically derived variational lower bound $L_{\text{vlb}}$ (in the case of $\Sigma(t) = \sigma_t^2 \mathbf{I}$) corresponds to the weighting scheme

$$w(t) = \frac{\beta_t^2}{2\sigma_t^2 \alpha_t (1 - \bar{\alpha}_t)} \tag{8}$$

with $\alpha_t$ and $\bar{\alpha}_t$ derived from the noise schedule $\beta_t$ (see Section 3). However, $L_{\text{vlb}}$ turns out to be extremely difficult to optimize in practice (see e.g., Dhariwal & Nichol (2021)), arguably due to large influence of the challenging denoising tasks near $t = 0$. To circumvent this issue, $L_{\text{simple}}$ was proposed by Ho et al. (2020) which corresponds to $w(t) = 1$, and turned out to be stable to optimize and already leads to remarkable perceptual quality. In order to achieve both, perceptual image quality and high likelihood values, Nichol & Dhariwal (2021) proposed $L_{\text{hybrid}} = L_{\text{simple}} + \lambda L_{\text{vlb}}$ with $\lambda = 0.001$ which increases the influence of low noise levels. The relative importance of reconstruction at different noise levels for the above discussed objectives is depicted in Figure 20.

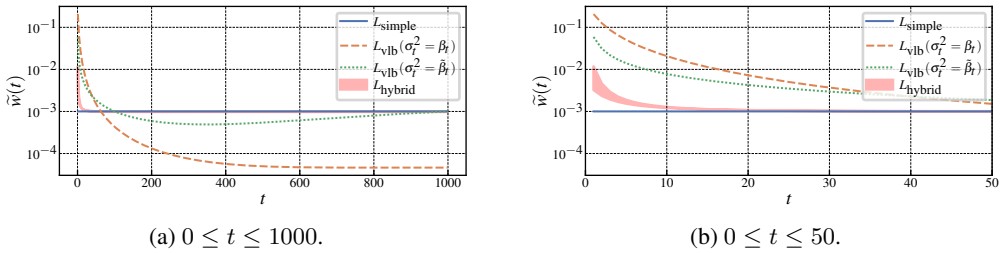

(a) $0 \le t \le 1000$.              (b) $0 \le t \le 50$.

Figure 20: **Relative importance of the reconstruction tasks for prominent DMs.** We show the relative influence $\widetilde{w}(t) = w(t)/\sum_{t=0}^{T} w(t)$ for (a) the whole denoising-diffusion process and (b) a close-up on the lowest noise levels. For $L_{\text{vlb}}$ we depict both, lower and upper bound for the denoising variance $\sigma_t^2$ (given as $\sigma_t^2 = \beta_t$ and $\sigma_t^2 = \tilde{\beta}_t$ as defined in Ho et al. (2020)). In the case of $L_{\text{hybrid}}$, used by IDDPM and ADM, we use $\sigma_t^2 \mathbf{I}$ for visualization purposes instead of a diagonal covariance matrix and plot $\widetilde{w}(t)$ for the range of admissible variances $\sigma_t^2$. All plots assume the linear noise schedule $\beta_t$ by Ho et al. (2020).

To put the above discussion in a nutshell, we hypothesize that the ability of a DM to match the real frequency spectrum is governed by the reconstruction performance at the corresponding noise levels. Importantly, successful error prediction at low noise scales, i.e., near $t = 0$, requires to capture the high-frequency content of an image. We deduce that the relative down-weighting of the influence of low noise levels when using $L_{\text{simple}}$ or $L_{\text{hybrid}}$ (compared to the theoretically derived $L_{\text{vlb}}$ which would account for high likelihoods) results in the observed mismatch of high frequencies.

The mean reduced spectra of various DMs (Figure 3) support this hypothesis: Both Nichol & Dhariwal (2021) and Dhariwal & Nichol (2021) train their model with $L_{\text{hybrid}}$ (which incorporates a relatively higher weight on low noise levels), and are able to reduce the gap to the real spectrum when compared to the baseline trained with $L_{\text{simple}}$ (Ho et al., 2020). Clearly, we believe that not only the weighting scheme $w(t)$ accounts for the resulting spectral properties, but more importantly the model's capabilities to successfully predict the low level noise in the first place. Nevertheless, the weighting scheme acts as a proxy that encourages the model to focus on specific noise levels.

We conclude that the objectives of DMs are well designed to guide the model to high perceptual quality (or benchmark metrics such as FID), while falling short on providing sufficient information to accurately model the high frequency content of the target images, which would be better captured by a likelihood-based objective like $L_{\text{vlb}}$.

# D    RESULTS ON ADDITIONAL DATASETS

To demonstrate the generalization of our findings, we perform classification and frequency analysis on additional data from ADM, PNDM, and LDM. Note that ADM-G-U refers to the two-stage up-sampling stack in which images are generated at a resolution of 64x64 and subsequently up-sampled to 256×256 pixels using a second model (Dhariwal & Nichol, 2021). The generated images are obtained according to the instructions given in Section A. Due to the relevance of facial images in the context of deepfakes, we also include two DMs not yet considered, P2 and ADM'(Choi et al., 2022), trained on FFHQ (Karras et al., 2019). ADM' is a smaller version of ADM with 93M instead of more than 500M parameters.[16] P2 is similar to ADM' but features a modified weighting scheme which improves performance by assigning higher weights to diffusion steps where perceptually rich contents are learned (Choi et al., 2022). We download checkpoints for both models from the official repository and sample images according to the authors' instructions.

Real images from LSUN (Yu et al., 2016), ImageNet (Russakovsky et al., 2015), and FFHQ (Karras et al., 2019) are downloaded from their official sources. Images from LSUN Cat/Horse, FFHQ, and ImageNet are resized and cropped to 256×256 pixels by applying the same pre-processing that was used when preparing the training data for the model they are compared against. All analyses are performed using 10k real and 10k generated images.

For results on other GANs, we refer to the original publications of the detectors (Wang et al., 2020; Gragnaniello et al., 2021; Mandelli et al., 2022a).

## D.1    PERFORMANCE OF UNIVERSAL DETECTORS

Similar to Table 2, we report AUROC, Pd@5%, and Pd@1% for the additional datasets in Table 6. Overall, the results do not invalidate the claim that universal GAN-detectors are incapable of effectively detecting images generated by DMs. Both PNDM and LDM trained on LSUN Church, however, are detected significantly better by all classifiers except Mandelli et al. (2022a). This might be caused by the relatively high FID (compared to the state of the art) both models achieve on this dataset (8.69 for PNDM, 4.02 for LDM).

Table 6: **Detection performance of pre-trained universal detectors on additional datasets.** For Wang et al. (2020) and Gragnaniello et al. (2021) we consider two different variants, respectively. The best score (determined by the highest Pd@1%) for each generator is highlighted in **bold**.

| AUROC / Pd@5% / Pd@1% | Wang et al. (2020) | | | | Gragnaniello et al. (2021) | | | | Mandelli et al. (2022a) | | |
|---|---|---|---|---|---|---|---|---|---|---|---|
| | Blur+JPEG (0.5) | | Blur+JPEG (0.1) | | ProGAN | | StyleGAN2 | | | | |
| ADM (LSUN Cat) | 58.4 / | 8.4 / | 2.5 | 58.1 / | 8.5 / | 3.3 | **60.2** / | **9.3** / | **4.2** | 51.7 / | 5.5 / | 1.8 | 55.6 / | 6.0 / | 1.3 |
| ADM (LSUN Horse) | 55.5 / | 6.7 / | 1.5 | 53.4 / | 6.0 / | 2.2 | **56.1** / | **7.5** / | **2.7** | 50.2 / | 4.8 / | 1.4 | 44.2 / | 2.4 / | 0.5 |
| ADM (ImageNet) | 69.1 / | 13.9 / | 4.1 | 71.7 / | 15.5 / | 4.5 | 72.1 / | 13.2 / | 3.5 | **83.9** / | **38.5** / | **16.6** | 60.1 / | 6.8 / | 0.9 |
| ADM-G-U (ImageNet) | 67.2 / | 11.7 / | 3.7 | 62.3 / | 6.2 / | 1.2 | 66.8 / | 6.0 / | 1.6 | **78.9** / | **26.7** / | **10.2** | 60.0 / | 7.6 / | 1.0 |
| PNDM (LSUN Church) | 76.9 / | 25.7 / | 10.2 | 77.6 / | 28.4 / | 12.0 | 90.9 / | 54.1 / | 24.5 | **99.3** / | **96.5** / | **85.8** | 56.4 / | 6.9 / | 1.9 |
| LDM (LSUN Church) | 86.3 / | 42.1 / | 19.8 | 82.2 / | 33.9 / | 14.2 | 98.8 / | 93.7 / | 75.5 | **99.5** / | **98.2** / | **90.2** | 58.9 / | 5.8 / | 1.3 |
| LDM (FFHQ) | 69.4 / | 14.2 / | 3.6 | 71.0 / | 14.8 / | 3.6 | **91.1** / | **54.2** / | **25.4** | 67.2 / | 10.2 / | 2.1 | 63.0 / | 5.6 / | 0.6 |
| ADM' (FFHQ) | 77.7 / | 24.7 / | 8.7 | 81.4 / | 28.2 / | 8.8 | **87.7** / | **41.9** / | **17.8** | 89.0 / | 45.5 / | 17.2 | 69.8 / | 11.1 / | 2.0 |
| P2 (FFHQ) | 79.5 / | 26.4 / | 8.9 | 83.2 / | 30.1 / | 9.2 | 89.2 / | 40.5 / | 11.5 | **91.1** / | **51.9** / | **18.9** | 72.5 / | 13.6 / | 2.7 |

## D.2    FREQUENCY ANALYSIS

We analyze the DFT (Figure 21), DCT (Figure 22), and reduced spectra (Figure 23) using a similar process as in Section 6.1. Regarding frequency artifacts, the results are consistent with that from LSUN Bedroom, LDM exhibits grid-like artifacts while ADM and PNDM do not. The spectra of ADM on LSUN Cat and LSUN Horse do contain irregular, vertical structures, which we did not observe for any other model and dataset. However, these are substantially different and not as pronounced as GAN artifacts. The DFT and DCT of (real and generated) FFHQ images clearly deviate from the remaining spectra, which we attribute to the homogeneity of the dataset. Note that LDM and ADM'/P2 were trained on differently processed versions of the real images, which is why we include the spectra of both variants. While we observe the known artifacts for LDM, ADM' and P2 do not contain such patterns.

---

[16] https://github.com/jychoi118/P2-weighting#training-your-models

The reduced spectra have largely the same characteristics as for LSUN Bedroom, with the exception of ImageNet. Here we observe an overestimation towards higher frequencies, which is the opposite of what we see for ADM on other datasets. A possible explanation could be that the authors sampled images from LSUN using 1000 and from ImageNet using only 250 steps. We suppose that the number of spectral discrepancies is highly training dependent.

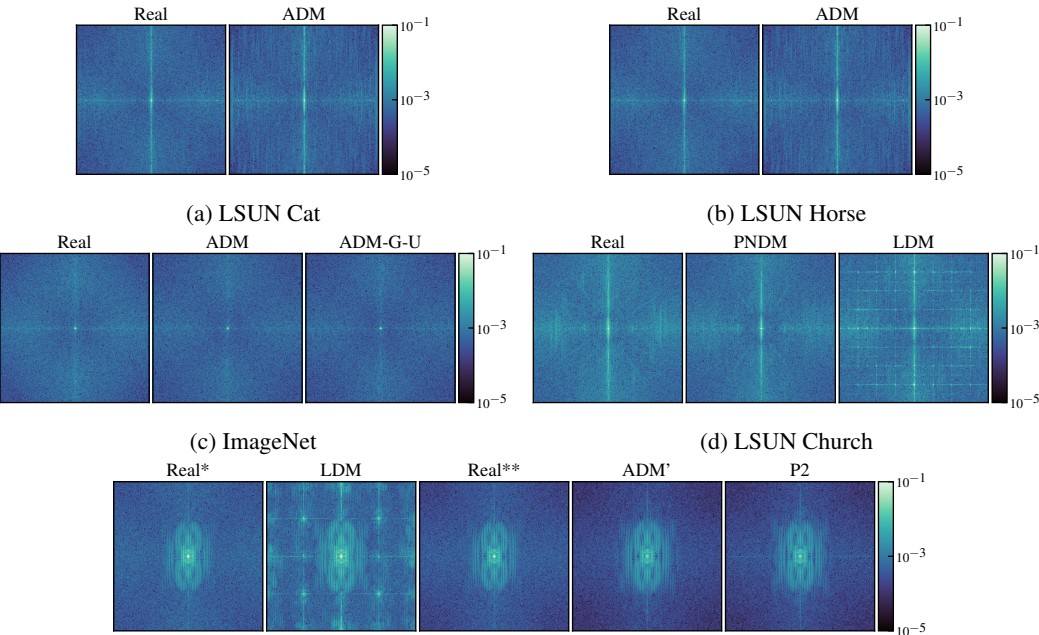

(a) LSUN Cat

(b) LSUN Horse

(c) ImageNet

(d) LSUN Church

(e) FFHQ (*: pre-processing according to Rombach et al. (2022), **: pre-processing according to Choi et al. (2022))

Figure 21: **Mean of DFT spectrum from real and generated images from additional datasets.** To increase visibility, the color bar is limited to $[10^{-5}, 10^{-1}]$, with values lying outside this interval being clipped.

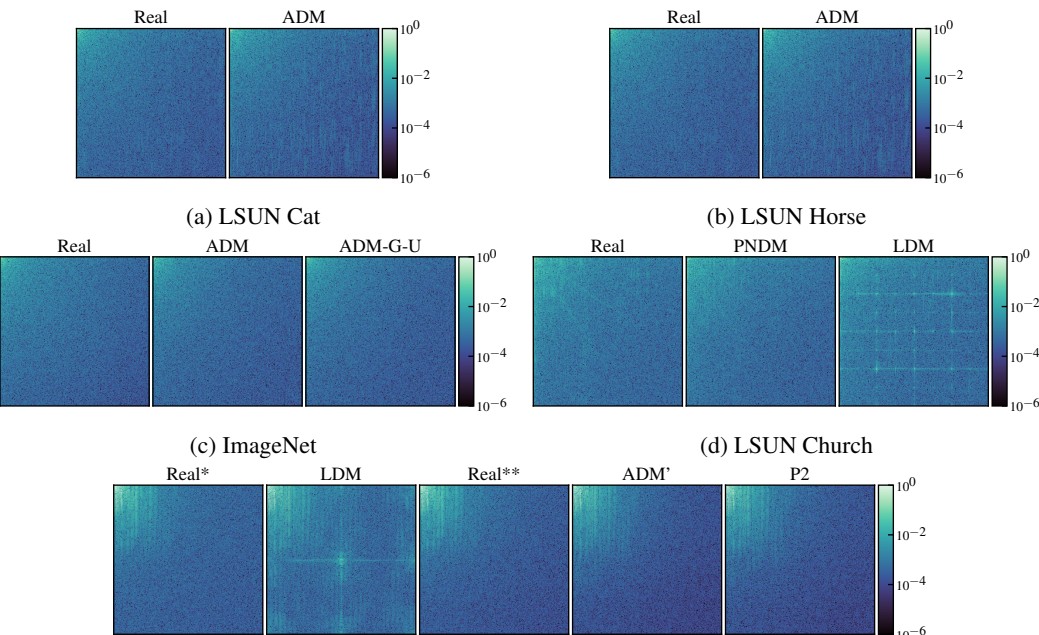

(e) FFHQ (*: pre-processing according to Rombach et al. (2022), **: pre-processing according to Choi et al. (2022))

Figure 22: **Mean of DCT spectrum from real and generated images from additional datasets.** To increase visibility, the color bar is limited to $[10^{-6}, 10^0]$, with values lying outside this interval being clipped.

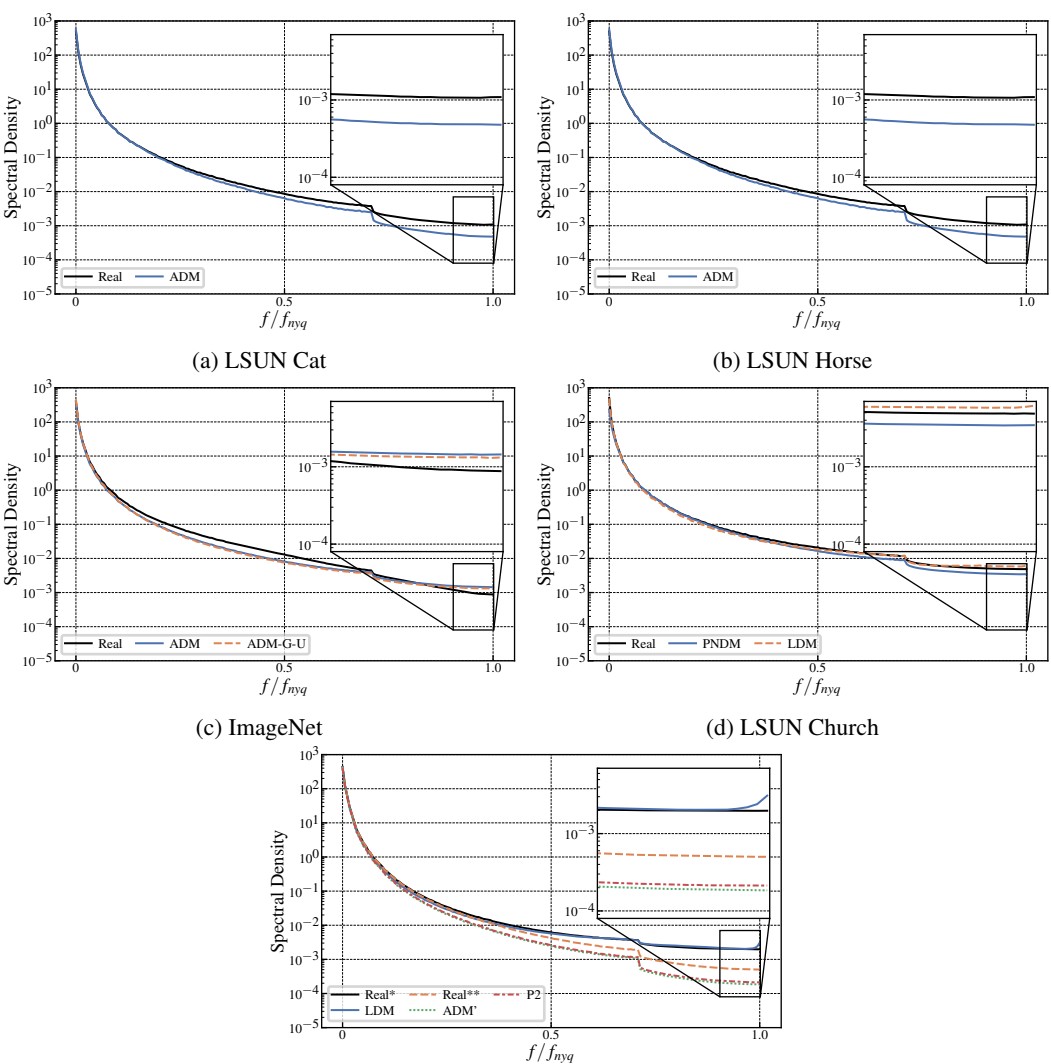

(a) LSUN Cat

(b) LSUN Horse

(c) ImageNet

(d) LSUN Church

(e) FFHQ (*: pre-processing according to Rombach et al. (2022), **: pre-processing according to Choi et al. (2022))

Figure 23: **Mean reduced spectrum from real and generated images from additional datasets.** The higher end of the spectrum where GAN-characteristic discrepancies occur is magnified.

