# OpenReview forum: "Towards the Detection of Diffusion Model Deepfakes"
_ICLR.cc/2023/Conference — Submitted to ICLR 2023_

### Official Review · Reviewer_rYBt · 2022-10-21

**Confidence:** 4
**Correctness:** 3
**Technical Novelty And Significance:** 2
**Empirical Novelty And Significance:** 4
**Recommendation:** 6

**Clarity, Quality, Novelty And Reproducibility:**

This work is very original in the experimentation part.
It has not the standard ICLR structure as it is more a refutation paper but it is a worthy work.

**Strength And Weaknesses:**

Strength:
- extensive experimentation on said state of the art deepfake detectors
- extensive experimentation on the influence of DM hyper parameters (sampling steps)
- discussion on the optimization target of these deepfake detectors

Weaknesses:
- need more maturation on a innovative methods to detect DM deepfakes

**Summary Of The Paper:**

This paper provides extensive experimentation on how state of the art deepfake detectors behave on diffusion model generated images.
They also propose an analysis in the frequency domain of diffusion model generated images compared to GAN and real images.


**Summary Of The Review:**

While the experiments and their interpretation are very interesting in trying state of the art detectors on DM fakes, this works lacks maturity in providing a novel approach to detect DM fakes.

---

> ### Author Response · Authors · 2022-11-11
> **Response to Reviewer rYBt**
>
> We thank the reviewer for their thoughtful feedback and are pleased that they recognize the originality of our work and the extensive experiments. We respond to their comment below.
>
> - *While the experiments and their interpretation are very interesting in trying state of the art detectors on DM fakes, this works lacks maturity in providing a novel approach to detect DM fakes.*
>
> We agree that the evaluation of existing, proven detectors is reasonable, which is why we selected this strategy to approach the novel problem of detecting DM-generated images. We also agree that the development of novel detection methods targeted at DMs is important. Therefore, one objective of our work is to enable this development by providing a solid foundation, especially regarding spectral properties, which in the past have shown to be a promising approach to detect generated images.

---

### Official Review · Reviewer_iMjn · 2022-10-25

**Confidence:** 2
**Correctness:** 3
**Technical Novelty And Significance:** 2
**Empirical Novelty And Significance:** 2
**Recommendation:** 5

**Clarity, Quality, Novelty And Reproducibility:**

The Clarity and Reproducibility of this paper are good. However, the Quality and Novelty are limited (see the above section).

**Strength And Weaknesses:**

Strengths:

1.	This paper focus on an ignored problem, the detection of DM-generated images.

2.	A benchmarking of the DM deepfake is conducted.

3.	This manuscript finds that the DM-generated images are different from GAN-generated images.


Weaknesses:

1.	There exist various generated models. However, this paper just analyzes the DM-generated and GAN-based models.

2.	This paper finds that existing detectors fail to effectively recognize DM-generated images and gives some suggestions. Unfortunately, they do not propose an effective method for deepfake detection.

3.	Only three detection methods are benchmarked in this paper. It is recommended to study more methods.


**Summary Of The Paper:**

In this paper, the authors focus on studying the detection of images generated by diffusion models. Specifically, this paper evaluates the performance of current methods and then analyzes the difference between GAN-generated and DM-generated images.

**Summary Of The Review:**

This paper focuses on an ignored problem. However, the novelty is limited and the experiments are inadequate.

---

> ### Author Response · Authors · 2022-11-11
> **Response to Reviewer iMjn**
>
> We are thankful for the reviewer's valuable feedback and encouraged that they point out the relevance of this work and clarity of the paper. We respond to their comments individually below.
>
> 1. *There exist various generated models. However, this paper just analyzes the DM-generated and GAN-based models.*
>
> While there are other classes of generative models, we consider GANs and DMs to be the most prominent in the context of high-quality image generation. In particular, the latter recently made a significant leap regarding text-to-image modeling. Moreover, our objective is to provide a first look at the detection of DM-generated images, with GANs serving as an adequate reference.
>
> 2. *This paper finds that existing detectors fail to effectively recognize DM-generated images and gives some suggestions. Unfortunately, they do not propose an effective method for deepfake detection.*
>
> We kindly point out that by finetuning the detector proposed in [1] to DM-generated images, it achieves near perfect performance on all evaluated datasets (see Figure 1).
>
> 3. *Only three detection methods are benchmarked in this paper. It is recommended to study more methods.*
>
> While a variety of works proposing methods to detect generated images exist, only very few claim to be 'universal', i.e., to perform well on data from unseen generators. The number of suitable methods is further reduced by missing or insufficient implementations, which led us to the final selection of the three state-of-the-art detectors. We also note that we consider two differently trained variants for two of the detection methods. We are happy to add additional methods if the reviewer has specific suggestions, any guidance is appreciated.
>
> - *This paper focuses on an ignored problem. However, the novelty is limited and the experiments are inadequate.*
>
> We are glad that the reviewer recognizes the relevance of this problem. We humbly emphasize that our work is the first to address the detection of DM-generated images, which, in our opinion, is much needed given the recent advances in text-to-image models based on DMs. In addition, our work provides (to the best of our knowledge) novel insights into their spectral characteristics, which in the past has proven to be an effective approach for the detection of GAN-generated images. Given our observation that frequency artifacts of DMs strongly deviate from those of GANs, we believe that our work can serve as a solid foundation for novel detection methods tailored to DM-generated images.
>
> Regarding your assessment of our experiments as 'inadequate' we would be grateful for specific suggestions on additional experiments. We kindly point towards the reviews of reviewers gmjk, 3UTf, and rYBt, which explicitly appreciate our experiments.
>
> ---
>
> [1] Wang, Sheng-Yu, Oliver Wang, Richard Zhang, Andrew Owens, and Alexei A. Efros. 2020. “CNN-Generated Images Are Surprisingly Easy to Spot... for Now.” In Proceedings of the IEEE/CVF Conference on Computer Vision and Pattern Recognition (CVPR).

---

### Official Review · Reviewer_3UTf · 2022-10-25

**Confidence:** 4
**Correctness:** 4
**Technical Novelty And Significance:** 3
**Empirical Novelty And Significance:** 3
**Recommendation:** 8

**Clarity, Quality, Novelty And Reproducibility:**

This paper is written well and experiments are sufficient. It investigates an area that has yet been explored by previous approaches, thus I think it is novel. Most models used in this approach are from official public resources, so reproducibility seems feasible.

**Strength And Weaknesses:**

The experimental setup of this paper is reasonable and adequate. The writing is clear and smooth. The results and conclusions are intuitive. I like how the authors present their results by asking questions and leading readers to answers.

I do not observe obvious drawbacks of this paper. It might make sense to include some results that can be easily classified as fake or misclassified from different approaches. This can intuitively show the features of those generated images that are confidently classified as fake.

**Summary Of The Paper:**

This paper presents an in-depth review of detection of diffusion-model generated images. This paper evaluates SOTA detectors on a range of DMs and then analyzes the differences of many aspects between DM-generated images and GAN generated images in addition to image fidelity.

**Summary Of The Review:**

I think the topic of this paper is meaningful and useful to a decent group of people. Conclusions are clear. And the experimental setup is well-rounded. In general, it is a good paper and I vote for acceptance.

---

> ### Author Response · Authors · 2022-11-11
> **Response to Reviewer 3UTf**
>
> We thank the reviewer for their thoughtful feedback. We are encouraged that they find our experiments suitable, our results intuitive, and agree with us on the novelty and relevance of our work. We respond to their comments below.
>
> - *It might make sense to include some results that can be easily classified as fake or misclassified from different approaches.*
>
> We agree that this is an interesting experiment and will add example images in different percentiles of the best-performing model's predictions to the revised paper. Our initial results confirm the findings in [1] (4.6) that the "fakeness" score of the detector does not necessarily correlate with visual quality. However, we observe that the colorfulness of an image appears to have an influence on the model output. We discuss this in more detail in the revision.
>
> ---
>
> [1] Wang, Sheng-Yu, Oliver Wang, Richard Zhang, Andrew Owens, and Alexei A. Efros. 2020. “CNN-Generated Images Are Surprisingly Easy to Spot... for Now.” In Proceedings of the IEEE/CVF Conference on Computer Vision and Pattern Recognition (CVPR).

---

### Official Review · Reviewer_Gcjc · 2022-10-27

**Confidence:** 5
**Correctness:** 3
**Technical Novelty And Significance:** 2
**Empirical Novelty And Significance:** 2
**Recommendation:** 5

**Clarity, Quality, Novelty And Reproducibility:**

The paper is easy to read and provides important information for a better understanding DM, and although the information is important, the paper does not provide any novelty.

**Details Of Ethics Concerns:**

none.

**Strength And Weaknesses:**

(+) The paper is an easy-to-read paper evaluating how well images generated by DM can be detected The paper evaluates the performance of existing deepfake detectors on images generated by DM. It reports performance results of finetuned existing deepfake detectors to DMs.
(-) The paper does not provide any novelty but instead provides more information regarding DM and compares detection results with GAN-generated images. The paper analyzes the spectral content of DM-generated images comparing GAN-generated images.



**Summary Of The Paper:**

This paper evaluates the performance of deepfake detectors on images generated by five diffusion models (DM) and five GANs. It also analyzes the spectral representation of DM-generated images.

**Summary Of The Review:**

The paper is easy to read and provides important information for a better understanding DM, and although the information is important, the paper does not provide any novelty.

---

> ### Author Response · Authors · 2022-11-11
> **Response to Reviewer Gcjc**
>
> We thank the reviewer for their valuable feedback. We are glad that they agree with us on the importance of obtaining a better understanding of DMs in the context of deepfake detection. We respond to the comments below.
>
> - *The paper does not provide any novelty but instead provides more information regarding DM and compares detection results with GAN-generated images.*
>
> We humbly emphasize that our work is the first to address the detection of DM-generated images, which, in our opinion, is much needed given the recent advances in text-to-image models based on DMs. In addition, our work provides (to the best of our knowledge) novel insights into their spectral characteristics, which in the past has proven to be an effective approach for the detection of GAN-generated images. Given our observation that frequency artifacts of DMs strongly deviate from those of GANs, we believe that our work can serve as a solid foundation for novel detection methods tailored to DM-generated images.

---

### Official Review · Reviewer_gmjk · 2022-10-27

**Confidence:** 5
**Correctness:** 3
**Technical Novelty And Significance:** 2
**Empirical Novelty And Significance:** 4
**Recommendation:** 6

**Clarity, Quality, Novelty And Reproducibility:**

- Clarity: Excellent. The paper is clearly presented.
- Quality: Average. The experiments and analysis are detailed and reasonable, but some further in-depth evaluations could be done.
- Novelty: Fair. The topic is new, but no novel methods are presented.
- Reproducibility: Excellent. Implementation details are described clearly, together with abundant supplementary materials.

**Strength And Weaknesses:**

Strengths:

1.  High-fidelity fake images generated by recent diffusion models have become a new threat in deepfake area. This paper presents a pioneer, systematic examination of this scenario with well-designed experiments.
2.  The authors provide several interesting observations from two different views: 1) the feasibility of detecting DM deepfakes using existing methods and 2) the characteristics of DM-generated images in frequency domain. These observations prove that DM deepfakes are more challenging to detect and have quite unique characteristics compared to GAN deepfakes, which brings useful insights to both deepfake detection and DM-based generation community.
3. The paper is overall well-written. All experiment details are presented clearly in the manuscript and supplementary materials with elaborate analysis.

Weaknesses:

Most of the main experiments are relatively simple and trivial. More in-depth analysis could be done to obtain richer and more complete conclusions. Specifically, I have the following suggestions:

1. In section 5.2, the authors make an interesting observation that detectors trained on DM-generated images can generalize better to GAN deepfakes than vice versa. They thus hypothesize that this is because detecting DM deepfakes is a more challenging task. Although the hypothesis seems to be supported by frequency domain analysis, a more intuitive interpretation might be achieved by further analysis on the feature space, e.g., a comparison between distributions of DM/GAN-generated features extracted by DM-trained detectors and GAN-trained detectors.

2. In section 6.2, the authors compare the effectiveness of detecting DM and GAN deepfakes in frequency domain using a simple logistic regression approach. However, some existing deepfake detection works absorb frequency info in a more 'soft' way, e.g. [1]. These frequency-aware deepfake methods' performance on DM deepfakes could also be examined. Will they perform better or worse on DM deepfakes than GAN deepfakes? Will the conclusion made in section 5.1 and 5.2 keep the same?

3. Most existing deepfake datasets (FF++, DFDC, DF1.0, etc.) mainly deal with forgery on human faces, which is an important and meaningful deepfake scenario. However in this paper, most conclusions are made on the LSUN bedroom dataset. The only face-relevant result (FFHQ) is presented in supplementary materials, but from Figure 18(e) we can observe a totally different pattern that contradicts the authors ' conclusion on high-frequency. Unique patterns of FFHQ can also be seen in Figure 16-17. I therefore suggest that the human face deepfake generated by DM should be discussed separately, and its uniqueness should be explained.

4. Since the training and inference of diffusion models are highly related to noise, the effect of perturbations widely used in deepfake datasets (Gaussian noise, blurring, compression, etc.) on DM deepfake detection might also be interesting to look at.


Reference

[1] Luo, Yuchen, et al. "Generalizing face forgery detection with high-frequency features." Proceedings of the IEEE/CVF conference on computer vision and pattern recognition. 2021.

**Summary Of The Paper:**

In this paper, the authors conduct an early study of the detection problem of diffusion model deepfakes. They prove that existing of-the-shelf deepfake detectors trained on GAN-generated images generalize badly to DM-generated deepfakes, while they can be effectively adapted to DM domain through fine-tuning. The authors further analyze the differences between the fake images generated by GAN and DM in the frequency domain, showing that DM-generated images exhibit unique characteristics.

**Summary Of The Review:**

This paper gives the first investigation into the deepfake detection problem on diffusion model-generated images. The authors present several meaningful insights into the unique properties of DM-generated deepfakes through well-designed experiments and detailed analysis. However, more in-depth discussions could be done to make the conclusions richer and more complete.

---

> ### Author Response · Authors · 2022-11-11
> **Response to Reviewer gmjk 1/2**
>
> We thank the reviewer for their valuable feedback and thoughtful suggestions! We are happy to see that they found our paper clear and well written, our experiments well designed, and the resulting insights useful. We respond to the comments individually below.
>
> 1. *In section 5.2, the authors make an interesting observation that detectors trained on DM-generated images can generalize better to GAN deepfakes than vice versa. They thus hypothesize that this is because detecting DM deepfakes is a more challenging task. Although the hypothesis seems to be supported by frequency domain analysis, a more intuitive interpretation might be achieved by further analysis on the feature space, e.g., a comparison between distributions of DM/GAN-generated features extracted by DM-trained detectors and GAN-trained detectors.*
>
> We strongly agree that gaining further insights based on the feature representations of deepfake detectors is insightful and thank the reviewer for suggesting this additional experiment, which we will add to the revised paper. We use the Maximum Mean Discrepancy [1] to compare the distributions of features extracted by the detection methods discussed in our paper. Besides the pre-trained detectors, we extend the analysis to the fine-tuned detection methods on DM-generated images, GAN-generated images, and both (as shown in Figure 1). The preliminary results of these additional experiments are in line with the presented hypothesis that DM-generated images are not as easily detectable as GAN-generated images. We will include the results in the revised version of the paper and add a discussion.
>
> 2. *In section 6.2, the authors compare the effectiveness of detecting DM and GAN deepfakes in frequency domain using a simple logistic regression approach. However, some existing deepfake detection works absorb frequency info in a more 'soft' way, e.g. [1]. These frequency-aware deepfake methods' performance on DM deepfakes could also be examined. Will they perform better or worse on DM deepfakes than GAN deepfakes? Will the conclusion made in section 5.1 and 5.2 keep the same?*
>
> We thank the reviewer for pointing us towards this work, which follows an interesting approach to detect face manipulations in videos by exploiting high-frequency noises. We note that it specifically targets video techniques like face swapping, which causes artifacts which are different from those of entirely artificially generated images. We therefore think this work is not fully suitable for our setting. We consciously opt for logistic regression because of its simplicity. We are aware that using a simple linear classifier is not the optimal choice for effectively detecting generated images, however, this is not the objective of this experiment. Similar to [4], our goal is to obtain a genuine comparison between the detection performance on pixel- and frequency data. We agree that for the experiments performed in Section 5, including a universal detector which specifically incorporates frequency information, might be interesting. We would be happy to include a suitable method suggested by the reviewer. However, as our results in Section 5.2 show, finetuning leads to almost perfect performance on both GANs and DMs, so there is little room for improvement by explicitly exploiting frequency information.

---

> ### Author Response · Authors · 2022-11-11
> **Response to Reviewer gmjk 2/2**
>
> 3. *Most existing deepfake datasets (FF++, DFDC, DF1.0, etc.) mainly deal with forgery on human faces, which is an important and meaningful deepfake scenario. However in this paper, most conclusions are made on the LSUN bedroom dataset. The only face-relevant result (FFHQ) is presented in supplementary materials, but from Figure 18(e) we can observe a totally different pattern that contradicts the authors' conclusion on high-frequency. Unique patterns of FFHQ can also be seen in Figure 16-17. I therefore suggest that the human face deepfake generated by DM should be discussed separately, and its uniqueness should be explained.*
>
> We agree that images of faces are particularly relevant in the context of generated images. In this work, as the reviewer also pointed out, our objective is to perform an initial, systematic evaluation of the detectability of DM-generated images. Therefore, as discussed in Section 4, we choose to analyze models which are all trained on the same dataset to avoid any biases caused by the choice of the dataset. We would have preferred to use a face dataset instead of LSUN Bedroom, but only few DMs offer checkpoints for FFHQ. Due to the importance of face images, we follow the reviewer's suggestion and add images generated by two additional DMs trained on FFHQ to the analysis in Appendix D. As they will be able to see in the revised paper, the frequency patterns and behavior towards high frequency are consistent with our findings for the remaining datasets. We attribute the seemingly contradictory results for the remaining generator, LDM, not to the dataset, but to the generator itself, which is adversarially trained and presumably therefore exhibits artifacts comparable to those of GANs.
>
> 4. *Since the training and inference of diffusion models are highly related to noise, the effect of perturbations widely used in deepfake datasets (Gaussian noise, blurring, compression, etc.) on DM deepfake detection might also be interesting to look at.*
>
> We agree with the reviewer's suggestion that an analysis of the influence of perturbations on the detection performance would provide interesting insights given the nature of DMs. We will add such an experiment in the revision. However, we note that two of the three evaluated detectors, namely those proposed in [2] and [3], do use data augmenation using blurring and compression during training.
>
> ---
>
> [1] Arthur Gretton, Karsten M. Borgwardt, Malte J. Rasch, Bernhard Schölkopf and Alexander Smola. 2012. "A Kernel Two-Sample Test" In JMLR 2012.
>
> [2] Wang, Sheng-Yu, Oliver Wang, Richard Zhang, Andrew Owens, and Alexei A. Efros. 2020. “CNN-Generated Images Are Surprisingly Easy to Spot... for Now.” In Proceedings of the IEEE/CVF Conference on Computer Vision and Pattern Recognition (CVPR).
>
> [3] Gragnaniello, D., D. Cozzolino, F. Marra, G. Poggi, and L. Verdoliva. 2021. “Are GAN Generated Images Easy to Detect? A Critical Analysis of the State-of-the-Art.” In 2021 IEEE International Conference on Multimedia and Expo (ICME), 1–6.
>
> [4] Frank, Joel, Thorsten Eisenhofer, Lea Schönherr, Asja Fischer, Dorothea Kolossa, and Thorsten Holz. 2020. “Leveraging Frequency Analysis for Deep Fake Image Recognition.” In Proceedings of the 37th International Conference on Machine Learning, 119:3247–58. Proceedings of Machine Learning Research. PMLR.

---

### Author Response · Authors · 2022-11-11
**Response to All Reviewers**

We thank all anonymous reviewers for their thoughtful feedback and helpful suggestions. We are currently in the process of performing the additional experiments we mention in the individual responses and preparing the revised paper. We intend to upload the updated version by the middle of next week to have further discussions based on the new results.

In the meantime, we would be grateful for further questions and suggestions.

---

> ### Author Response · Authors · 2022-11-16
> **Revision Uploaded**
>
> We just submitted the revised paper with additional experiments.
>
> The updated version contains the following additions:
> - We perform a feature space analysis using MMD and t-SNE to gain further insights into the detectability of GAN- and DM-generated images in Appendix B.3 (based on a suggestion by Reviewer gmjk). The new results support our previous findings and allow us to refine our hypothesis according to which DM-generated images are more difficult to detect.
> - We study the effect of common image perturbations (blur, crop, JPEG compression, noise) on the detector's performance in Appendix B.4 (based on a suggestion by Reviewer gmjk).
> - We provide example images which the detectors consider more or less fake, based on their output value in Appendix B.5 (based on a suggestion by Reviewer 3UTf).
> - We add two DMs trained on FFHQ to the evaluation of additional datasets in Appendix D and address the deviating frequency spectra (based on a suggestion by Reviewer gmjk).
>
> The aforementioned changes are highlighted in blue to increase their visibility. In addition, we fixed some typos and rephrased some sentences. These do not alter any statements of our work and are therefore not highlighted.
>
> We look forward to discussing the new results and would be pleased to answer further questions.

---

### Decision · Program_Chairs · 2023-01-20

**Decision:**

Reject

**Justification For Why Not Higher Score:**

The observations and assumptions in this paper might not be generalizable across different visual domains. Therefore, comprehensive investigations are needed to make the findings more convincing.


**Justification For Why Not Lower Score:**

N/A

**Metareview: Summary, Strengths And Weaknesses:**

This paper was reviewed by three experts in the field. The AC and reviewers had a thorough discussion on this paper, where the reviewers raised many concerns regarding the paper: 1) The observations and assumptions in this paper might not be generalizable across different domains (e.g., faces, indoor, and outdoor). 2) The experiments are not comprehensive enough regarding the various types of diffusion models. Considering the reviewers' concerns, we regret that the paper cannot be recommended for acceptance at this time. The authors are encouraged to consider the reviewers' comments when revising the paper for submission elsewhere.


**Summary Of Ac-Reviewer Meeting:**

During the AC-reviewer discussion, the reviewers raised two major concerns regarding the paper: 1) The observations and assumptions in this paper might not be generalizable across different domains (e.g., faces, indoor, and outdoor). 2) The experiments are not comprehensive enough regarding the various types of diffusion models.